# How Powerful are $K$-hop Message Passing Graph Neural Networks

**Jiarui Feng**[1,2]  **Yixin Chen**[1]  **Fuhai Li**[2]  **Anindya Sarkar**[1]  **Muhan Zhang**[3,4]
{feng.jiarui, fuhai.li, anindya}@wustl.edu,
chen@cse.wustl.edu, muhan@pku.edu.cn
[1]Department of CSE, Washington University in St. Louis
[2]Institute for Informatics, Washington University School of Medicine
[3]Institute for Artificial Intelligence, Peking University
[4]Beijing Institute for General Artificial Intelligence

## Abstract

The most popular design paradigm for Graph Neural Networks (GNNs) is 1-hop message passing—aggregating information from 1-hop neighbors repeatedly. However, the expressive power of 1-hop message passing is bounded by the Weisfeiler-Lehman (1-WL) test. Recently, researchers extended 1-hop message passing to $K$-hop message passing by aggregating information from $K$-hop neighbors of nodes simultaneously. However, there is no work on analyzing the expressive power of $K$-hop message passing. In this work, we theoretically characterize the expressive power of $K$-hop message passing. Specifically, we first formally differentiate two different kernels of $K$-hop message passing which are often misused in previous works. We then characterize the expressive power of $K$-hop message passing by showing that it is more powerful than 1-WL and can distinguish almost all regular graphs. Despite the higher expressive power, we show that $K$-hop message passing still cannot distinguish some simple regular graphs and its expressive power is bounded by 3-WL. To further enhance its expressive power, we introduce a KP-GNN framework, which improves $K$-hop message passing by leveraging the peripheral subgraph information in each hop. We show that KP-GNN can distinguish many distance regular graphs which could not be distinguished by previous distance encoding or 3-WL methods. Experimental results verify the expressive power and effectiveness of KP-GNN. KP-GNN achieves competitive results across all benchmark datasets.

## 1 Introduction

Currently, most existing graph neural networks (GNNs) follow the *message passing* framework, which iteratively aggregates information from the neighbors and updates the representations of nodes. It has shown superior performance on graph-related tasks [1, 2, 3, 4, 5, 6, 7] comparing to traditional graph embedding techniques [8, 9]. However, as the procedure of message passing is similar to the 1-dimensional Weisfeiler-Lehman (1-WL) test [10], the expressive power of message passing GNNs is also bounded by the 1-WL test [7, 11]. Namely, GNNs cannot distinguish two non-isomorphic graph structures if the 1-WL test fails.

In normal message passing GNNs, the node representation is updated by the direct neighbors of the node, which are called 1-hop neighbors. Recently, some works extend the notion of message passing into $K$-hop message passing [12, 13, 14, 15, 16]. $K$**-hop message passing** is a type of message passing where the node representation is updated by aggregating information from not only 1st hop but all the neighbors within $K$ hops of the node. However, there is no work on theoretically

36th Conference on Neural Information Processing Systems (NeurIPS 2022).

characterizing the expressive power of GNNs with $K$-hop message passing, e.g., whether it can improve the 1-hop message passing or not and to what extent it can.

In this work, we theoretically characterize the expressive power of $K$-hop message passing GNNs. Specifically, 1) we formally distinguish two different kernels of the $K$-hop neighbors, which are often misused in previous works. The first kernel is based on whether the node can be reached within $k$ steps of the graph diffusion process, which is used in GPR-GNN [15] and MixHop [12]. The second one is based on the shortest path distance of $k$, which is used in GINE+ [16] and Graphormer [17]. Further, we show that different kernels of $K$-hop neighbors will result in **different expressive power** of $K$-hop message passing. 2) We show that $K$**-hop message passing is strictly more powerful than 1-hop message passing and can distinguish almost all regular graphs. 3) However, it still failed in distinguishing some simple regular graphs, no matter which kernel is used, and its expressive power is bounded by 3-WL**. This motivates us to improve $K$-hop message passing further.

Here, we introduce **KP-GNN**, a new GNN framework with $K$-hop message passing, which significantly improves the expressive power of standard $K$-hop message passing GNNs. In particular, during the aggregation of neighbors in each hop, KP-GNN not only aggregates neighboring nodes in that hop but also aggregates the **peripheral subgraph** (subgraph induced by the neighbors in that hop). This additional information helps the KP-GNN to learn more expressive local structural features around the node. We further show that KP-GNN can distinguish many distance regular graphs with a proper encoder for the peripheral subgraph. The proposed KP-GNN has several additional advantages. First, it can be applied to most existing $K$-hop message-passing GNNs with only slight modification. Second, it only adds little computational complexity to standard $K$-hop message passing. We demonstrate the effectiveness of the KP-GNN framework through extensive experiments on both simulation and real-world datasets.

## 2  $K$-hop message passing and its expressive power

### 2.1  Notations

Denote a graph as $G = (V, E)$, where $V = \{1, 2, ..., n\}$ is the node set and $E \subseteq V \times V$ is the edge set. Meanwhile, denote $A \in \{0, 1\}^{n \times n}$ as the adjacency matrix of graph $G$. Denote $x_v$ as the feature vector of node $v$ and denote $e_{uv}$ as the feature vector of the edge from $u$ to $v$. Finally, we denote $Q^1_{v,G}$ as the set of 1-hop neighbors of node $v$ in graph $G$ and $\mathcal{N}^1_{v,G} = Q^1_{v,G} \cup \{v\}$. Note that when we say $K$-hop neighbors of node $v$, we mean **all** the neighbors that have distance from node $v$ less than or equal to $K$. In contrast, $k$-th hop neighbors mean the neighbors with **exactly** distance $k$ from node $v$. The definition of distance will be discussed in section 2.3.

### 2.2  1-hop message passing framework

Currently, most existing GNNs are designed based on 1-hop message passing framework [18]. Denote $h^l_v$ as the output representation of node $v$ at layer $l$ and $h^0_v = x_v$. Briefly, given a graph $G$ and a 1-hop message passing GNN, at layer $l$ of the GNN, $h^l_v$ is computed by $h^{l-1}_v$ and $\{\!\{h^{l-1}_u \mid u \in Q^1_{v,G}\}\!\}$:

$$m^l_v = \text{MES}^l(\{\!\{(h^{l-1}_u, e_{uv}) | u \in Q^1_{v,G}\}\!\}), \quad h^l_v = \text{UPD}^l(m^l_v, h^{l-1}_v), \tag{1}$$

where $m^l_v$ is the message to node $v$ at layer $l$, $\text{MES}^l$ and $\text{UPD}^l$ are message and update functions at layer $l$ respectively. After $L$ layers of message passing, $h^L_v$ is used as the final representation of node $v$. Such a representation can be used to conduct node-level tasks like node classification and node regression. To get the graph representation, a readout function is used:

$$h_G = \text{READOUT}(\{\!\{h^L_v | v \in V\}\!\}), \tag{2}$$

where READOUT is the readout function for computing the final graph representation. Then $h_G$ can be used to conduct graph-level tasks like graph classification and graph regression.

### 2.3  $K$-hop message passing framework

The 1-hop message passing framework can be directly generalized to $K$-hop message passing, as it shares the same message and update mechanism. The difference is that independent message

and update functions can be employed for each hop. Meanwhile, a combination function is needed to combine the results from different hops into the final node representation at this layer. First, we differentiate two different kernels of $K$-hop neighbors, which are interchanged and misused in previous research.

The first kernel of $K$-hop neighbors is *shortest path distance (spd) kernel*. Namely, the $k$-th hop neighbors of node $v$ in graph $G$ is the set of nodes with the shortest path distance of $k$ from $v$.

**Definition 1.** *For a node $v$ in graph $G$, the $K$-hop neighbors $\mathcal{N}_{v,G}^{K,spd}$ of $v$ based on shortest path distance kernel is the set of nodes that have the shortest path distance from node $v$ **less than or equal to** $K$. We further denote $Q_{v,G}^{k,spd}$ as the set of nodes in $G$ that are **exactly** the $k$-th hop neighbors (with shortest path distance of exactly $k$) and $\mathcal{N}_{v,G}^{0,spd} = Q_{v,G}^{0,spd} = \{v\}$ is the node itself.*

The second kernel of the $K$-hop neighbors is based on *graph diffusion (gd)*.

**Definition 2.** *For a node $v$ in graph $G$, the $K$-hop neighbors $\mathcal{N}_{v,G}^{K,gd}$ of $v$ based on graph diffusion kernel is the set of nodes that can diffuse information to node $v$ **within the number of random walk diffusion steps** $K$ and the diffusion kernel $A$ (adjacency matrix). We further denote $Q_{v,G}^{k,gd}$ as the set of nodes in $G$ that are **exactly** the $k$-th hop neighbors (nodes that can diffuse information to node $v$ with $k$ diffusion steps) and $\mathcal{N}_{v,G}^{0,gd} = Q_{v,G}^{0,gd} = \{v\}$ is the node itself.*

Note that a node can be a $k$-th hop neighbor of $v$ for multiple $k$ based on the graph diffusion kernel, but it can only appear in one hop for the shortest path distance kernel. We include more discussions of $K$-hop kernels in Appendix A. Next, we define the $K$-hop message passing framework as follows:

$$m_v^{l,k} = \text{MES}_k^l(\{\!\{(h_u^{l-1}, e_{uv})|u \in Q_{v,G}^{k,t})\}\!\}), \quad h_v^{l,k} = \text{UPD}_k^l(m_v^{l,k}, h_v^{l-1}),$$
$$h_v^l = \text{COMBINE}^l(\{\!\{h_v^{l,k}|k = 1, 2, ..., K\}\!\}), \tag{3}$$

where $t \in \{spd, gd\}$, $spd$ is the shortest path distance kernel and $gd$ is the graph diffusion kernel. Here, for each hop, we can apply unique MES and UPD functions. Note that for $k > 1$, there may not exist the edge feature $e_{uv}$ as nodes are not directly connected. But we leave it here since we can use other types of features to replace it like path encoding. We further discuss it in Appendix I. Compared to the 1-hop message passing framework described in Equation (1), the COMBINE function is introduced to combine the representations of node $v$ at different hops. It is easy to see that a $L$ layer 1-hop message passing GNNs is actually a $L$ layer $K$-hop message passing GNNs with $K = 1$. We include more discussions of $K$-hop message passing GNNs in Appendix A. To aid further analysis, we also prove that $K$-hop message passing can injectively encode the neighbor representations at different hops into $h_v^l$ in Appendix B.

## 2.4 Expressive power of $K$-hop message passing framework

In this section, we theoretically analyze the expressive power of $K$-hop message passing. We assume there is no edge feature and all nodes in the graph have the same feature, which means that GNNs can only distinguish two nodes with the local structure of nodes. Note that including node features only increases the expressive power of GNNs as nodes/graphs are more easily to be discriminated. It has been proved that the expressive power of 1-hop message passing is bounded by the 1-WL test on discriminating non-isomorphic graphs [7, 11]. In this section, We show that the $K$-hop message passing is strictly more powerful than the 1-WL test when $K > 1$. Across the analysis, we utilize regular graphs as examples to illustrate our theorems since they cannot be distinguished using either 1-hop message passing or the 1-WL test. To begin the analysis, we first define *proper $K$-hop message passing GNNs*.

**Definition 3.** *A proper $K$-hop message passing GNN is a GNN model where the message, update, and combine functions are all injective given the input from a countable space.*

A proper $K$-hop message passing GNN is easy to find due to the universal approximation theorem [19] of neural network and the Deep Set for set operation [20]. In the latter sections, by default, all mentioned $K$-hop message passing GNNs are proper. Next, we introduce *node configuration*:

**Definition 4.** *The node configuration of node $v$ in graph $G$ within $K$ hops under $t$ kernel is a list $A_{v,G}^{K,t} = (a_{v,G}^{1,t}, a_{v,G}^{2,t}, ..., a_{v,G}^{K,t})$, where $a_{v,G}^{i,t} = |Q_{v,G}^{i,t}|$ is the number of $i$-th hop neighbors of node $v$.*

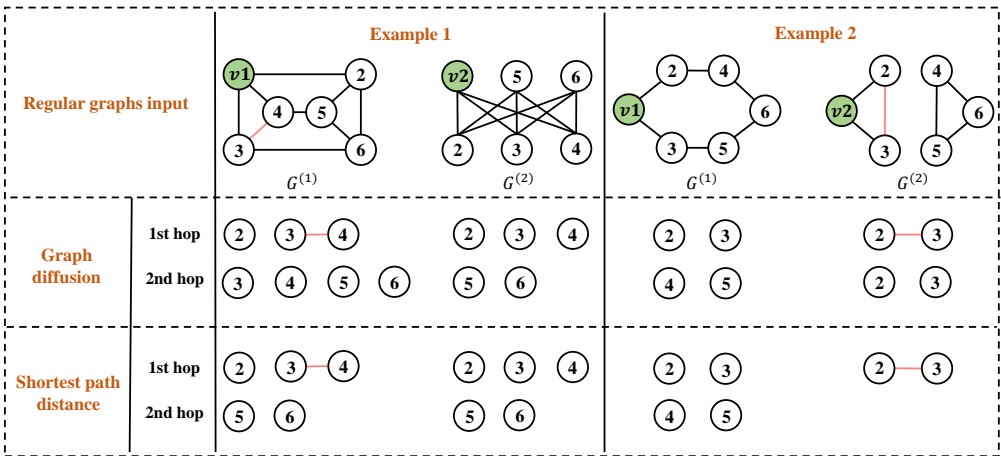

Figure 1: Here are two pairs of non-isomorphic regular graphs. With 2-hop message passing, example 1 can be distinguished by the graph diffusion kernel, and example 2 can be distinguished by the shortest path distance kernel. However, both two examples become indistinguishable if we switch the kernel. Finally, both two examples can be distinguished by adding peripheral subgraph information.

When we say two node configurations $A^{K,t}_{v_1,G^{(1)}}$ and $A^{K,t}_{v_2,G^{(2)}}$ are equal, we mean that these two lists are component-wise equal to each other. Now, we state the first proposition:

**Proposition 1.** *A proper $K$-hop message passing GNN is strictly more powerful than 1-hop message passing GNNs when $K > 1$.*

To see why this is true, we first discuss how node configuration relates to the first layer of message passing. In the first layer of $K$-hop message passing, each node aggregates neighbors from up to $K$ hops. As each node has the same node label, an injective message function can only know how many neighbors at each hop, which is exactly the node configuration. In other words, **The first layer of $K$-hop message passing is equivalent to inject node configuration to each node label**. When $K = 1$, the node configuration of $v_1$ and $v_2$ are $d_{v_1,G^{(1)}}$ and $d_{v_2,G^{(2)}}$, where $d_{v,G}$ is the node degree of $v$. After $L$ layers, GNNs can only get the node degree information of each node within $L$ hops of node $v$. Then, it is straightforward to see why these GNNs cannot distinguish any $n$-sized $r$-regular graph, as each node in the regular graph has the same degree.

Next, when $K > 1$, the $K$-hop message passing is at least equally powerful as 1-hop message passing since node configuration up to $K$ hop includes all the information the 1 hop has. To see why it is more powerful, we use two examples to illustrate it. The first example is shown in the left part of Figure 1. Suppose here we use graph diffusion kernel and we want to learn the representation of node $v_1$ and node $v_2$ in the two graphs. We know that the 1-hop GNNs produce the same representation for two nodes as they are both nodes in 6-sized 3-regular graphs. However, it is easy to see that $v_1$ and $v_2$ have different local structures and should have different representations. Instead, if we use the 2-hop message passing with the graph diffusion kernel, we can easily distinguish two nodes by checking the 2nd hop neighbors of the node, as node $v_1$ has four 2nd hop neighbors but node $v_2$ only has two 2nd hop neighbors. The second example is shown in the right part of Figure 1. Two graphs in the example are still regular graphs. Suppose here we use shortest path distance kernel, node $v_1$ and $v_2$ have different numbers of 2nd hop neighbors and thus will have different representations by performing 2-hop message passing. These two examples convincingly demonstrate that the $K$-hop message passing with $K > 1$ can have better expressive power than $K = 1$. To further study the expressive power of $K$-hop message passing on regular graphs, we show the following result:

**Theorem 1.** *Consider all pairs of $n$-sized $r$-regular graphs, let $3 \leq r < (2log2n)^{1/2}$ and $\epsilon$ be a fixed constant. With at most $K = \lfloor (\frac{1}{2} + \epsilon) \frac{\log 2n}{\log (r-1)} \rfloor$, there exists a 1 layer $K$-hop message passing GNN using the shortest path distance kernel that distinguishes almost all $1 - o(n^{-1/2})$ such pairs of graphs.*

We include the proof and simulation results in Appendix C. Theorem 1 shows that even with 1 layer and a modest $K$, $K$-hop GNNs are powerful enough to distinguish almost all regular graphs.

Finally, we characterize the existing $K$-hop methods with the proposed $K$-hop message passing framework. Specifically, we show that 1) the expressive power of $K$ layer GINE [16] is bounded by $K$ layer $K$-hop message passing with the shortest path distance kernel. 2) The expressive power of the Graphormer [17] is equal to $K$-hop GNNs with the shortest path distance kernel and infinity $K$. 3) For spectral GNNs and existing $K$-hop GNNs with the graph diffusion kernel like MixHop [12] and MAGNA [14], we find they actually use a **weak version of $K$-hop than the definition of us**. Specifically, it is shown that the expressive power of spectral GNNs is also bounded by 1-WL test [21], which contradicts our result as graph diffusion can be viewed as a special case of spectral GNN. However, we show that our definition of $K$-hop message passing with graph diffusion kernel actually injects a non-linear function on the spectral basis, thus achieving superior expressive power. We leave the detailed discussion in Appendix D. Further, Distance Encoding [22] also uses the shortest path distance information to augment the 1-hop message passing, which is similar to $K$-hop GNNs with the shortest path distance kernel. However, we find the expressive power of the two frameworks differs from each other. We leave the detailed discussion in Appendix E.

## 2.5 Limitation of $K$-hop message passing framework

Although we show that $K$-hop GNNs with $K > 1$ are better at distinguishing non-isomorphic structures than 1-hop GNNs, there are still limitations. In this section, we discuss the limitation of $K$-hop message passing. Specifically, we show that **the choice of the kernel can affect the expressive power of $K$-hop message passing**. Furthermore, even with $K$-hop message passing, we still cannot distinguish some simple non-isomorphic structures and **the expressive power of $K$-hop message passing is bounded by 3-WL**.

Continue looking at the provided examples in Figure 1. In example 1, if we use the shortest path distance kernel instead of the graph diffusion kernel, two nodes have the same number of neighbors in the 2nd hop, which means that we cannot distinguish two nodes this time. Similarly, in example 2, two nodes have the same number of neighbors in both 1st and 2nd hops using graph diffusion kernel. These results highlight that the choice of the kernel can affect the expressive power of $K$-hop message passing, and none of them can distinguish both two examples with 2-hop message passing.

Recently, Frasca et al. [23] show that any subgraph-based GNNs with node-based selection policy can be implemented by 3-IGN [24, 25] and thus their expressive power is bounded by 3-WL test. Here, we show that the $K$-hop message passing GNNs can also be implemented by 3-IGN for both two kernels and thus:

**Theorem 2.** *The expressive power of a proper $K$-hop message passing GNN of any kernel is bounded by the 3-WL test.*

We include the proof in Appendix F. Given all these observations, we may wonder if there is a way to further improve the expressive power of $K$-hop message passing?

# 3 KP-GNN: improving the power of $K$-hop message passing by peripheral subgraph

In this section, we describe how to improve the expressive power of $K$-hop message passing by adding additional information to the message passing framework. Specifically, by adding *peripheral subgraph* information, we can improve the expressive power of the $K$-hop message passing by a large margin.

## 3.1 Peripheral edge and peripheral subgraph

First, we define *peripheral edge* and *peripheral subgraph*.

**Definition 5.** *The peripheral edge $E(Q_{v,G}^{k,t})$ is defined as the set of edges that connect nodes within set $Q_{v,G}^{k,t}$. We further denote $|E(Q_{v,G}^{k,t})|$ as the number of peripheral edge in $E(Q_{v,G}^{k,t})$. The peripheral subgraph $G_{v,G}^{k,t} = (Q_{v,G}^{k,t}, E(Q_{v,G}^{k,t}))$ is defined as the subgraph induced by $Q_{v,G}^{k,t}$ from the whole graph $G$.*

Briefly speaking, the peripheral edge $E(Q_{v,G}^{k,t})$ record all the edges whose two ends are both from $Q_{v,G}^{k,t}$ and the peripheral subgraph is a graph constituted by peripheral edges. It is easy to see that the peripheral subgraph $G_{v,G}^{k,t}$ automatically contains all the information of peripheral edge $E(Q_{v,G}^{k,t})$. Next, we show that the power of $K$-hop message passing can be improved by leveraging the information of peripheral edges and peripheral subgraphs. We again refer to the examples in Figure 1. Here we only consider the peripheral edge information. In example 1, we notice that at the 1st hop, there is an edge between node 3 and node 4 in the left graph. More specifically, $E(Q_{v_1,G^{(1)}}^{1,t}) = \{(3,4)\}$. In contrast, we have $E(Q_{v_2,G^{(2)}}^{1,t}) = \{\}$ in the right graph, which means there is no edge between the 1st hop neighbors of $v_2$. Therefore, we can successfully distinguish these two nodes by adding this information to the message passing. Similarly, in example 2, there is one edge between the 1st hop neighbors of node $v_2$, but no such edge exists for node $v_1$. By leveraging peripheral edge information, we can also distinguish the two nodes. The above examples demonstrate the effectiveness of the peripheral edge and peripheral subgraph information.

## 3.2 $K$-hop peripheral-subgraph-enhanced graph neural network

In this section, we propose **K**-hop **P**eripheral-subgraph-enhanced **G**raph **N**eural **N**etwork (KP-GNN), which equips $K$-hop message passing GNNs with peripheral subgraph information for more powerful GNN design. Recall the $K$-hop message passing defined in Equation (3). The only difference between KP-GNN and original $K$-hop GNNs is that we revise the message function as follows:

$$m_v^{l,k} = \text{MES}_k^l(\{\!\{(h_u^{l-1}, e_{uv})|u \in Q_{v,G}^{k,t}\}\!\}, G_{v,G}^{k,t}). \tag{4}$$

Briefly speaking, in the message step at the $k$-th hop, we not only aggregate information of the neighbors but also the peripheral subgraph at that hop. The implementation of KP-GNN can be very flexible, as any graph encoding function can be used. To maximize the information the model can encode while keeping it simple, we implement the message function as:

$$\begin{aligned} \text{MES}_k^l = {} & \text{MES}_k^{l,normal}(\{\!\{(h_u^{l-1}, e_{uv})|u \in Q_{v,G}^{k,t}\}\!\}) + f(G_{v,G}^{k,t}), \\ f(G_{v,G}^{k,t}) = {} & \text{EMB}((E(Q_{v,G}^{k,t}), C_k^{k'})) , \end{aligned} \tag{5}$$

where $\text{MES}_k^{l,normal}$ denotes the message function in the original GNN model, $C_k^{k'}$ is the $k'$-configuration, which encode both node configuration and the number of the peripheral edge of all nodes in $G_{v,G}^{k,t}$ up to $k'$ hops. It can be regarded as running another 1 layer KP-GNN and readout function on each peripheral subgraph. EMB is a learnable embedding function. With this implementation, any base GNN model can be incorporated into and be enhanced by the KP-GNN framework by replacing $\text{MES}_k^{l,normal}$ and $\text{UPD}_k^l$ with the corresponding functions for each hop $k$. We leave the detailed implementation in Appendix I.

## 3.3 The expressive power of KP-GNN and comparison with existing methods

In this section, we theoretically characterize the expressive power of KP-GNN and compare it with the original $K$-hop message passing framework. The key insight is that, according to Equation (4), the message function at the $k$-th hop additionally encodes $G_{v,G}^{k,t}$ compared to normal $K$-hop message passing. As we have already shown in the last section, $K$-hop GNNs are bounded by 3-WL and thus cannot distinguish any non-isomorphic distance regular graphs, as well as Distance Encoding [22]. Let $C_{j,G}^{k'}$ be the $k'$-configuration of peripheral subgraph at $j$-th hop of nodes in distance regular graph $G$. Here we show that with the aid of peripheral subgraphs, KP-GNN is able to distinguish distance regular graphs:

**Proposition 2.** *For two non-isomorphic distance regular graphs $G^{(1)} = (V^{(1)}, E^{(1)})$ and $G^{(2)} = (V^{(2)}, E^{(2)})$ with the same diameter $d$ and intersection array $(b_0, b_1, ..., b_{d-1}; c_1, c_2, ..., c_d)$. Given a proper 1-layer $d$-hop KP-GNN with message functions defined in Equation (5), it can distinguish $G^{(1)}$ and $G^{(2)}$ if $C_{j,G^{(1)}}^{k'} \neq C_{j,G^{(2)}}^{k'}$ for some $0 < j \leq d$.*

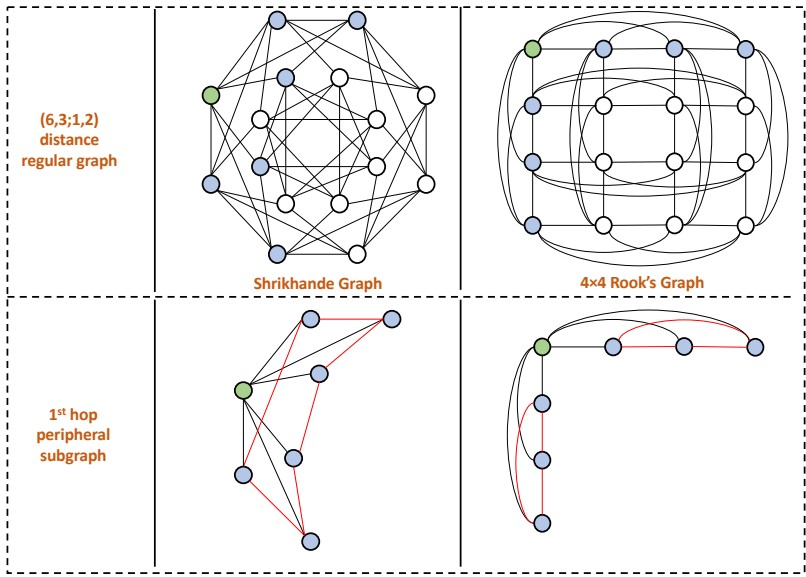

Figure 2: An example of two non-isomorphic distance regular graph with intersection array $(6, 3; 1, 2)$.

We include the proof in Appendix G. Here we leverage an example in Figure 2 to briefly show why KP-GNN is able to distinguish distance regular graphs. Figure 2 displays two distance regular graphs with an intersection array of $(6, 3; 1, 2)$. The left one is Shrikhande graph and the right one is $4\times4$ Rook's graph. Now, let's look at the 1-hop peripheral subgraph of the green node. In the Shrikhande graph, there are 6 peripheral edges marked with red. Further, 6 edges constitute a circle. In the $4\times4$ Rook's graph, there are still 6 peripheral edges. However, 6 edges constitute two circles with 3 edges in each circle, which is different from the Shrikhande graph. Then, any peripheral subgraph encoder that can distinguish these two graphs like node configuration enables the corresponding KP-GNN to distinguish the example. Proposition 2 shows that the KP-GNN is capable of distinguishing distance regular graphs, which further distinguishes KP-GNN from DE-1 [22] as it cannot distinguish any two connected distance regular graphs with the same intersection arrays according to Theorem 3.7 in [22]. However, it is currently unknown whether can KP-GNN with Equation 5 distinguish all distance regular graphs. We leave the detailed discussion in Appendix G.

Moreover, both the subgraph-based GNNs like NGNN [26], GNN-AK [27], ESAN [28], and KP-GNN leverage the information in the subgraph to enhance the power of message passing. However, KP-GNN is intrinsically different from them. Firstly, in KP-GNN, the message passing is performed on the whole graph instead of the subgraphs. This means that for each node, there is only one representation to be learned. Instead, for subgraph-based GNNs, the message passing is performed separately for each subgraph and each node could have multiple representations depending on which subgraph it is in. Secondly, in subgraph-based GNNs, they consider the subgraph as a whole without distinguishing nodes at different hops. Instead, KP-GNN takes one step further by dividing the subgraph into two parts. The first part is the hierarchy of neighbors at each hop. The second part is the connection structure between nodes in each hop. This gives us a better point of view to design a more powerful learning method. From the Corollary 7 in [23], we know that all subgraph-based GNNs with node selection as subgraph policy is bounded by 3-WL, which means they cannot distinguish any distance regular graph and KP-GNN is better at it.

## 3.4 Time, space complexity, and limitation

In this section, we discuss the time and space complexity of $K$-hop message passing GNN and KP-GNN. Suppose a graph has $n$ nodes and $m$ edges. Then, the $K$-hop message passing and KP-GNN have both the space complexity of $O(n)$ and the time complexity of $O(n^2)$ for the shortest path distance kernel. Note that the complexity of graph diffusion is no less than the shortest path distance kernel. **We can see that KP-GNN only requires the same space complexity as vanilla GNNs and much less time complexity than the subgraph-based GNNs, which are at least $O(nm)$.** However, $K$-hop message passing including KP-GNN still have intrinsic limitation. We leave a detailed discussion on the complexity and limitation of KP-GNN in Appendix H.

# 4 Related Work

**Expressive power of GNN.** Analyzing the expressive power of GNNs is a crucial problem as it can serve as a guide on how to improve GNNs. Xu et al. [7] and Morris et al. [11] first proved that the power of 1-hop message passing is bounded by the 1-WL test. In other words, 1-hop message passing cannot distinguish any non-isomorphic graphs that the 1-WL test fails to. In recent years, many efforts have been put into increasing the expressive power of 1-hop messaging passing. The first line of research tries to mimic the higher-order WL tests, like 1-2-3 GNN [11], PPGN [24], ring-GNN [29]. However, they require exponentially increasing space and time complexity w.r.t. node number and cannot be generalized to large-scale graphs. The second line of research tries to enhance the rooted subtree of 1-WL with additional features. Some works [30, 31, 32] add one-hot or random features into nodes. Although they achieve good results in some settings, they deteriorate the generalization ability as such features produce different representations for nodes even with the same local graph structure. Some works like Distance Encoding [22], SEAL [33], labeling trick [34] and GLASS [35] introduce node labeling based on either distance or distinguishing target node set. On the other hand, GraphSNN [36] introduces a hierarchy of local isomorphism and proposes structural coefficients as additional features to identify such local isomorphism. However, the function designed to approximate the structural coefficient cannot fully achieve its theoretical power. The third line of research resorts to subgraph representation. Specifically, ID-GNN [37] extracts ego-netwok for each node and labels the root node with a different color. NGNN [26] encodes a rooted subgraph instead of a rooted subtree by subgraph pooling thus achieving superior expressive power on distinguishing regular graphs. GNN-AK [27] applies a similar idea as NGNN. The only difference lies in how to compute the node representation from the local subgraph. However, such methods need to run an inner GNN on every node of the graph thus introducing much more computation overhead. Meanwhile, the expressive power of subgraph GNNs are bounded by 3-WL [23].

$K$**-hop message passing GNN.** There are some existing works that instantiate the $K$-hop message passing framework. For example, MixHop [12] performs message passing on each hop with graph diffusion kernel and concatenates the representation on each hop as the final representation. K-hop [13] sequentially performs the message passing from hop K to hop 1 to compute the representation of the center node. However, it is not parallelizable due to its computational procedure. MAGNA [14] introduces an attention mechanism to $K$-hop message passing. GPR-GNN [15] use graph diffusion kernel to perform graph convolution on $K$-hop and aggregate them with learnable parameters. However, none of them give a formal definition of $K$-hop message passing and theoretically analyze its representation power and limitations.

# 5 Experiments

In this section, we conduct extensive experiments to evaluate the performance of KP-GNN. Specifically, we 1) empirically verify the expressive power of KP-GNN on 3 simulation datasets and demonstrate the benefits of KP-GNN compared to normal $K$-hop message passing GNNs; 2) demonstrate the effectiveness of KP-GNN on identifying various node properties, graph properties, and substructures with 3 simulation datasets; 3) show that the KP-GNN can achieve state-of-the-art performance on multiple real-world datasets; 4) analyze the running time of KP-GNN. The detail of each variant of KP-GNN is described in Appendix I and the detailed experimental setting is described in Appendix J. We implement the KP-GNN with PyTorch Geometric package [38]. Our code is available at `https://github.com/JiaruiFeng/KP-GNN`.

**Datasets**: To evaluate the expressive power of KP-GNN, we choose: 1) EXP dataset [31], which contains 600 pairs of non-isomorphic graphs (1-WL failed). The goal is to map these graphs to two different classes. 2) SR25 dataset [39], which contains 15 non-isomorphic strongly regular graphs (3-WL failed) with each graph of 25 nodes. The dataset is translated to a 15-way classification problem with the goal of mapping each graph into different classes. 3) CSL dataset [40], which contains 150 4-regular graphs (1-WL failed) divided into 10 isomorphism classes. The goal of the task is to classify them into corresponding isomorphism classes. To demonstrate the capacity of KP-GNN on counting node/graph properties and substructures, we pick 1) Graph property regression (connectedness, diameter, radius) and node property regression (single source shortest path, eccentricity, Laplacian feature) task on random graph dataset [41]. 2) Graph substructure counting (triangle, tailed triangle, star, and 4-cycle) tasks on random graph dataset [42]. To evaluate the performance of KP-GNN on

real-world datasets, we select 1) MUTAG [43], D&D [44], PROTEINS [44], PTC-MR [45], and IMDB-B [46] from TU database. 2) QM9 [47, 48] and ZINC [49] for molecular properties prediction. The detailed statistics of the datasets are described in Appendix L. Without further highlighting, all error bars in the result tables are the standard deviations of multiple runs.

Table 1: Empirical evaluation of the expressive power.

| Method | K | EXP (ACC) | | SR (ACC) | | CSL (ACC) | |
|--------|---|-----------|-----------|----------|-----------|-----------|-----------|
| | | SPD | GD | SPD | GD | SPD | GD |
| **K-GIN** | K=1 | 50 | 50 | 6.67 | 6.67 | 12 | 12 |
| | K=2 | 50 | 50 | 6.67 | 6.67 | 32 | 22.7 |
| | K=3 | 100 | 66.9 | 6.67 | 6.67 | 62 | 42 |
| | K=4 | 100 | 100 | 6.67 | 6.67 | 92.7 | 62.7 |
| **KP-GIN** | K=1 | 50 | 50 | 100 | 100 | 22 | 22 |
| | K=2 | 100 | 100 | 100 | 100 | 52.7 | 52.7 |
| | K=3 | 100 | 100 | 100 | 100 | 90 | 90 |
| | K=4 | 100 | 100 | 100 | 100 | 100 | 100 |

**Empirical evaluation of the expressive power**: For empirical evaluation of the expressive power, we conduct the ablation study on hop $K$ for both normal $K$-hop GNNs and KP-GNN. For $K$-hop GNNs, we implement K-GIN which uses GIN [7] as the base encoder. For KP-GNN, we implement KP-GIN. The results are shown in Table 1. Based on the results, we have the following conclusions: 1) $K$-hop GNNs with both two kernels have expressive power higher than the 1-WL test as it shows the perfect performance on the EXP dataset and performance better than a random guess on the CSL dataset. 2) Increasing $K$ can improve the expressive for both two kernels. 3) $K$-hop GNNs cannot distinguish any strong regular graphs in SR25 dataset, which is aligned with Theorem 2. 4) KP-GNN has much higher expressive power than normal $K$-hop GNNs by showing better performance on every dataset given the same $K$. Further, it achieves perfect results on the SR25 dataset even with $K = 1$, which demonstrates its ability on distinguishing distance regular graphs.

Table 2: Simulation dataset result. The top two are highlighted by **First**, **Second**.

| Method | Node Properties ($\log_{10}$(MSE)) | | | Graph Properties ($\log_{10}$(MSE)) | | | Counting Substructures (MAE) | | | |
|--------|------|------|------|----------|----------|--------|------|------------|------|---------|
| | SSSP | Ecc. | Lap. | Connect. | Diameter | Radius | Tri. | Tailed Tri. | Star | 4-Cycle |
| **GIN** | -2.0000 | -1.9000 | -1.6000 | -1.9239 | -3.3079 | -4.7584 | 0.3569 | 0.2373 | 0.0224 | 0.2185 |
| **PNA** | **-2.8900** | **-2.8900** | -3.7700 | -1.9395 | 3.4382 | -4.9470 | 0.3532 | 0.2648 | 0.1278 | 0.2430 |
| **PPGN** | - | - | - | -1.9804 | -3.6147 | -5.0878 | 0.0089 | 0.0096 | **0.0148** | **0.0090** |
| **GIN-AK+** | - | - | - | **-2.7513** | **-3.9687** | -5.1846 | 0.0123 | 0.0112 | **0.0150** | **0.0126** |
| **K-GIN+** | -2.7919 | -2.5938 | **-4.6360** | -2.1782 | **-3.9695** | -5.3088 | 0.2593 | 0.1930 | 0.0165 | 0.2079 |
| **KP-GIN+** | -2.7969 | -2.6169 | **-4.7687** | **-4.4322** | -3.9361 | **-5.3345** | **0.0060** | **0.0073** | 0.0151 | 0.0395 |

**Effectiveness on node/graph properties and substructure prediction**: To evaluate the effectiveness of KP-GNN on node/graph properties and substructure prediction, we compare it with several existing models. For the baseline model, we use GIN [7], which has the same expressive power as the 1-WL test. For more powerful baselines, we use GIN-AK+ [27], PNA [41], and PPGN [24]. For normal $K$-hop GNNs, we implement K-GIN+, and for KP-GNN, we implement KP-GIN+. The results are shown in Table 2. Baseline results are taken from [27] and [41]. We can see KP-GIN+ achieve SOTA on a majority of tasks. Meanwhile, K-GIN+ also gets great performance on node/graph properties prediction. These results demonstrate the capability of KP-GNN to identify various properties and substructures. We leave the detailed results on counting substructures in Appendix K

Table 3: TU dataset evaluation result.

| Method | MUTAG | D&D | PTC-MR | PROTEINS | IMDB-B |
|--------|-------|-----|--------|----------|--------|
| **WL** | 90.4±5.7 | 79.4±0.3 | 59.9±4.3 | 75.0±3.1 | 73.8±3.9 |
| **GIN** | 89.4±5.6 | - | 64.6±7.0 | 75.9±2.8 | 75.1±5.1 |
| **DGCNN** | 85.8±1.7 | 79.3 ±0.9 | 58.6 ±2.5 | 75.5±0.9 | 70.0±0.9 |
| **GraphSNN** | 91.24±2.5 | 82.46±2.7 | 66.96±3.5 | 76.51±2.5 | 76.93±3.3 |
| **GIN-AK+** | 91.30±7.0 | - | 68.20±5.6 | 77.10±5.7 | 75.60±3.7 |
| **KP-GCN** | 91.7±6.0 | 79.0±4.7 | 67.1±6.3 | 75.8±3.5 | 75.9±3.8 |
| **KP-GraphSAGE** | 91.7±6.5 | 78.1±2.6 | 66.5±4.0 | 76.5±4.6 | 76.4±2.7 |
| **KP-GIN** | 92.2±6.5 | 79.4±3.8 | 66.8±6.8 | 75.8±4.6 | 76.6±4.2 |
| **GIN-AK+**[*] | 95.0±6.1 | OOM | 74.1±5.9 | 78.9±5.4 | 77.3±3.1 |
| **GraphSNN**[*] | 94.70±1.9 | **83.93±2.3** | 70.58±3.1 | 78.42±2.7 | 78.51±2.8 |
| **KP-GCN**[*] | **96.1±4.6** | 83.2±2.2 | **77.1±4.1** | 80.3±4.2 | 79.6±2.5 |
| **KP-GraphSAGE**[*] | **96.1±4.6** | 83.6±2.4 | 76.2±4.5 | **80.4±4.3** | 80.3±2.4 |
| **KP-GIN**[*] | 95.6±4.4 | 83.5±2.2 | 76.2±4.5 | 79.5±4.4 | **80.7±2.6** |

Table 4: QM9 results. The top two are highlighted by **First**, **Second**.

| Target | DTNN | MPNN | Deep LRP | PPGN | N-1-2-3-GNN | KP-GIN+ | KP-GIN$'$ |
|---|---|---|---|---|---|---|---|
| $\mu$ | **0.244** | 0.358 | 0.364 | **0.231** | 0.433 | 0.367 | 0.358 |
| $\alpha$ | 0.95 | 0.89 | 0.298 | 0.382 | 0.265 | **0.242** | **0.233** |
| $\varepsilon_{\text{HOMO}}$ | 0.00388 | 0.00541 | 0.00254 | 0.00276 | 0.00279 | **0.00247** | **0.00240** |
| $\varepsilon_{\text{LUMO}}$ | 0.00512 | 0.00623 | 0.00277 | 0.00287 | 0.00276 | **0.00238** | **0.00236** |
| $\Delta\varepsilon$ | 0.0112 | 0.0066 | 0.00353 | 0.00406 | 0.00390 | **0.00345** | **0.00333** |
| $\langle R^2 \rangle$ | 17.0 | 28.5 | 19.3 | 16.7 | 20.1 | **16.49** | **16.51** |
| ZPVE | 0.00172 | 0.00216 | 0.00055 | 0.00064 | **0.00015** | 0.00018 | **0.00017** |
| $U_0$ | 2.43 | 2.05 | 0.413 | 0.234 | 0.205 | **0.0728** | **0.0682** |
| $U$ | 2.43 | 2.00 | 0.413 | 0.234 | 0.200 | **0.0553** | 0.0696 |
| $H$ | 2.43 | 2.02 | 0.413 | 0.229 | 0.249 | **0.0575** | 0.0641 |
| $G$ | 2.43 | 2.02 | 0.413 | 0.238 | 0.253 | 0.0526 | **0.0484** |
| $C_v$ | 0.27 | 0.42 | 0.129 | 0.184 | **0.0811** | 0.0973 | **0.0869** |

**Evaluation on TU datasets**: For baseline models, we select: 1) graph kernel-based method: WL subtree kernel [50]; 2) vanilla GNN methods: GIN [7] and DGCNN [6]; 3) advanced GNN methods: GraphSNN [36] and GIN-AK+ [27]. For the proposed KP-GNN, we implement GCN [1], GraphSAGE [3], and GIN [7] using the KP-GNN framework, denoted as KP-GCN, KP-GraphSAGE, and KP-GIN respectively. The results are shown in Table 3. For a more fair and comprehensive comparison, we report the results from two different evaluation settings. The first setting follows Xu et al. [7] and the second setting follows Wijesinghe and Wang [36]. We denote the second setting with $^*$ in the table. We can see KP-GNN achieves SOTA performance on most of datasets under the second setting and still comparable performance to other baselines under the first setting.

Table 5: ZINC result.

| Method | # param. | test MAE |
|---|---|---|
| **MPNN** | 480805 | 0.145±0.007 |
| **PNA** | 387155 | 0.142±0.010 |
| **Graphormer** | 489321 | 0.122±0.006 |
| **GSN** | ~500000 | 0.101±0.010 |
| **GIN-AK+** | - | 0.080±0.001 |
| **CIN** | - | **0.079±0.006** |
| **KP-GIN+** | 499099 | 0.111±0.006 |
| **KP-GIN$'$** | 488649 | 0.093±0.007 |

**Evaluation on molecular prediction tasks**: For QM9 dataset, we report baseline results of DTNN and MPNN from [48]. We further select Deep LRP [42], PPGN [24], and Nested 1-2-3-GNN [26] as baseline models. For the ZINC dataset, we report results of MPNN [18] and PNA [41] from [17]. We further pick Graphormer [17], GSN [51], GIN-AK+ [27], and CIN [52]. For KP-GNN, we choose KP-GIN+ and KP-GIN$'$. The results of the QM9 dataset are shown in Table 4. We can see KP-GNN achieves SOTA performance on most of the targets. The results of the ZINC dataset are shown in Table 5. Although KP-GNN does not achieve the best result, it is still comparable to other methods.

Table 6: Running time (s/epoch).

| Method | D&D | ZINC | Graph property |
|---|---|---|---|
| **GIN** | 1.10 | 3.59 | 1.02 |
| **K-GIN** | 3.94 | 6.44 | 1.67 |
| **KP-GIN** | 4.19 | 7.38 | 1.94 |
| **KP-GIN+** | 4.28 | 6.74 | 1.93 |

**Running time comparison**: In this section, we compare the running time of KP-GNN to 1-hop message passing GNN and $K$-hop message passing GNN. We use GIN [7] as the base model. We also include the KP-GIN+. All models use the same number of layers and hidden dimensions for a fair comparison. The results are shown in Table 6. We set $K = 4$ for all datasets. We can see the computational overhead is almost linear to $K$. This is reasonable as practical graphs are sparse and the number of $K$-hop neighbors is far less than $n$ when using a small $K$.

## 6 Conclusion

In this paper, we theoretically characterize the power of $K$-hop message passing GNNs and propose the KP-GNN to improve the expressive power by leveraging the peripheral subgraph information at each hop. Theoretically, we prove that $K$-hop GNNs can distinguish almost all regular graphs but are bounded by the 3-WL test. KP-GNN is able to distinguish many distance regular graphs. Empirically, KP-GNN achieves competitive results across all simulation and real-world datasets.

## 7 Acknowledgement

This work is partially supported by NSF grant CBE-2225809 and NSF China (No. 62276003).

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
