# A  More about the $K$-hop kernel and $K$-hop message passing

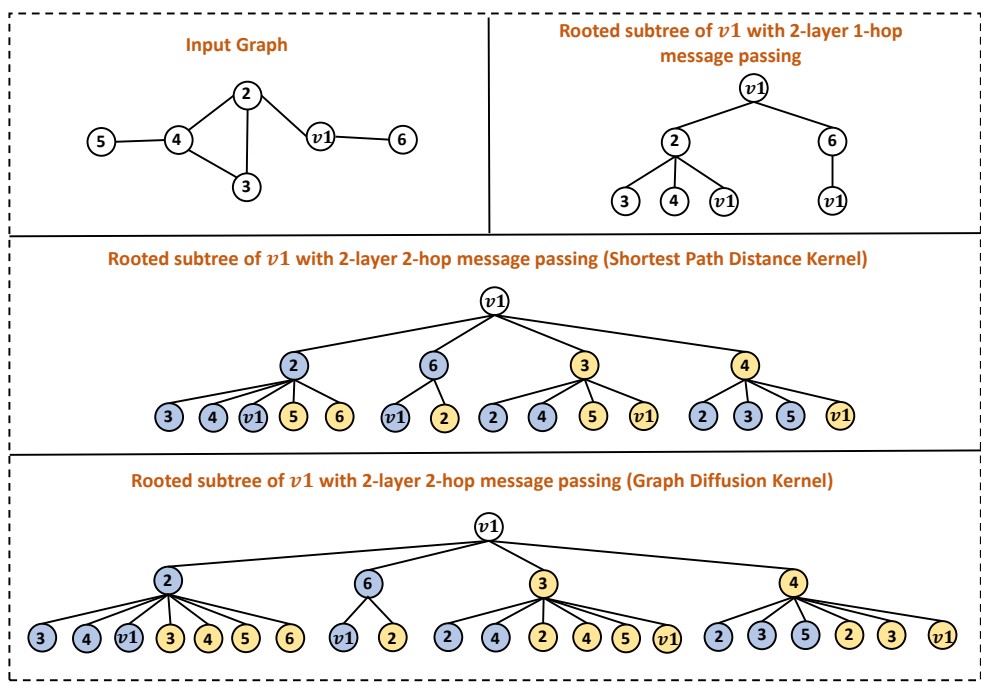

Figure 3: The rooted subtree of node $v1$ with 1-hop message passing and $K$-hop message passing. Here we assume that $K = 2$ and the number of layers is 2.

In this section, we further discuss two different types of K-hop kernel and K-hop message passing.

## A.1  More about $K$-hop kernel

First, recall the shortest path distance kernel and graph diffusion kernel defined in Definition 1 and 2. Given two definitions, the first thing we can conclude is that the $K$-hop neighbors of node $v$ under two different kernels will be the same, namely $\mathcal{N}_{v,G}^{K,spd} = \mathcal{N}_{v,G}^{K,gd}$ as both two kernels capture all nodes that can be reached from node $v$ within the distance of $K$. Second, we have $\mathcal{N}_{v,G}^{1,spd} = Q_{v,G}^{1,spd} = \mathcal{N}_{v,G}^{1,gd} = Q_{v,G}^{1,gd}$, which means the neighbor set is same for both the shortest path distance kernel and the graph diffusion kernel when $K = 1$. The third thing is that $Q_{v,G}^{k,spd}$ will not always equal to $Q_{v,G}^{k,gd}$ for some $k$. Since for shortest path distance kernel, one node will only appear in at most one of $Q_{v,G}^{k,spd}$ for $k = 1, 2, ..., K$. Instead, nodes can appear in multiple $Q_{v,G}^{k,gd}$. This is the key reason why the choice of the kernel can affect the expressive power of $K$-hop message passing.

## A.2  More about $K$-hop message passing

Here, we use an example shown in Figure 3 to illustrate how K-hop message passing works and compare it with 1-hop message passing. The input graph is shown on the left top of the figure. Suppose we want to learn the representation of node $v1$ using 2 layer message passing GNNs. First, if we perform 1-hop message passing, it will encode a 2-height rooted subtree, which is shown on the right top of the figure. Note that each node is learned using the same set of parameters, which is indicated by filling each node with the same color (white in the figure). Now, we consider performing a 2-hop message passing GNN with the shortest path distance kernel. The rooted subtree of node $v1$ is shown in the middle of the figure. we can see that at each height, both 1st hop neighbors and 2nd hop neighbors are included. Furthermore, different sets of parameters are used for different hops, which is indicated by filling nodes in the different hops with different colors (blue for 1st hop and yellow for 2nd hop). Finally, at the bottom of the figure, we show the 2-hop message passing GNN with graph diffusion kernel. It is easy to see the rooted subtree is different from the one that uses the shortest path distance kernel, as nodes can appear in both the 1st hop and 2nd hop of neighbors.

# B  Proof of injectiveness of Equation (3)

In this section, we formally prove that Equation (3) is an injective mapping of the neighbor representations at different hops. As here each layer $l$ is doing exactly the same procedure, we only need to prove it for one iteration. Therefore we ignore the superscript and rewrite Equation (3) as:

$$m_v^k = \text{MES}_k(\{\!\!\{(h_u, e_{uv})|u \in Q_{v,G}^{k,t}\}\!\!\}), \quad h_v^k = \text{UPD}_k(m_v^k, h_v),$$
$$\hat{h}_v = \text{COMBINE}(\{\!\!\{h_v^k|k = 1, 2, ..., K\}\!\!\}). \tag{6}$$

Next, we state the following proposition:

**Proposition 3.** *There exist injective functions $MES_k, UPD_k, k = 1, 2, ..., K$, and an injective multiset function COMBINE, such that $\hat{h}_v$ is an injective mapping of $\{\!\!\{(k, \{\!\!\{(h_u, e_{uv})|u \in Q_{v,G}^{k,t}\}\!\!\}, h_v) \mid k = 1, 2, ..., K\}\!\!\}$.*

*Proof.* The existence of injective message passing ($\text{MES}_k, \text{UPD}_k$) and multiset pooling (COMBINE) functions are well proved in [7]. So below we prove the injectiveness of $\hat{h}_v$. First, we combine $\text{MSE}_k$ and $\text{UPD}_k$ together into $\phi_k$:

$$h_v^k = \phi_k(\{\!\!\{(h_u, e_{uv})|u \in Q_{v,G}^{k,t}\}\!\!\}, h_v). \tag{7}$$

Note that $\phi_k$ is still injective as the composition of injective functions is injective. Next, we need to prove that $h_v^k$ is an injective mapping of $(k, \{\!\!\{(h_u, e_{uv})|u \in Q_{v,G}^{k,t}\}\!\!\}, h_v)$. To prove it, we rewrite the function $\phi_k(\cdot)$ into $\phi(k)(\cdot)$, that is, $\phi$ is an injective function taking $k$ as input and outputs a function $\phi_k = \phi(k)$. We let $\phi(k_1)(x_1)$ and $\phi(k_2)(x_2)$ output different values for $k_1 \neq k_2$ given any input $x_1, x_2$, e.g., always let the final output dimension be $k$. Then, we can rewrite $h_v^k$ as

$$h_v^k = \phi(k)(\{\!\!\{(h_u, e_{uv})|u \in Q_{v,G}^{k,t}\}\!\!\}, h_v)$$
$$= \psi(k, \{\!\!\{(h_u, e_{uv})|u \in Q_{v,G}^{k,t}\}\!\!\}, h_v), \tag{8}$$

where we have composed two injective functions $\phi(\cdot)$ and $\phi(k)(\cdot)$ into a single one $\psi(\cdot)$. Since given different $k$, $\phi(k)(\cdot)$ always outputs distinct values for any input $k$, and given fixed $k$, $\phi(k)(\cdot)$ always outputs different values for different $(\{\!\!\{(h_u, e_{uv})|u \in Q_{v,G}^{k,t}\}\!\!\}, h_v)$, the resulting $\psi(\cdot)$ always outputs different values for different $(k, \{\!\!\{(h_u, e_{uv})|u \in Q_{v,G}^{k,t}\}\!\!\}, h_v)$, i.e., it injectively maps $(k, \{\!\!\{(h_u, e_{uv})|u \in Q_{v,G}^{k,t}\}\!\!\}, h_v)$. Thus, we have proved that $h_v^k$ is an injective mapping of $(k, \{\!\!\{(h_u, e_{uv})|u \in Q_{v,G}^{k,t}\}\!\!\}, h_v)$.

Finally, since COMBINE is an injective multiset function, we conclude that its output $\hat{h}_v$ is an injective mapping of $\{\!\!\{(k, \{\!\!\{(h_u, e_{uv})|u \in Q_{v,G}^{k,t}\}\!\!\}, h_v) \mid k = 1, 2, ..., K\}\!\!\}$. $\qquad\square$

# C  Proof of Theorem 1 and simulation result

## C.1  Proof of Theorem 1

Here we restate Theorem 1: Consider all pairs of $n$-sized $r$-regular graphs, let $3 \leq r < (2log2n)^{1/2}$ and $\epsilon$ be a fixed constant. With at most $K = \lfloor (\frac{1}{2} + \epsilon)\frac{\log 2n}{\log (r-1)} \rfloor$, there exists a 1 layer $K$-hop message passing GNN using the shortest path distance kernel that distinguishes almost all $1 - o(n^{-1/2})$ such pairs of graphs.

As we state in the main paper, 1 layer $K$-hop GNN is equivalent to inject node configuration for each node label. Therefore, it is sufficient to show that given node configuration $A_{v,G}^{K,spd}$ for all $v \in V$, we can distinguish almost every pair of regular graphs. First, we introduce the following lemma.

**Lemma 1.** *For two graphs $G^{(1)} = (V^{(1)}, E^{(1)})$ and $G^{(2)} = (V^{(2)}, E^{(2)})$ that are randomly and independently sampled from $n$-sized $r$-regular graphs with $3 \leq r < (2\log 2n)^{1/2}$. We pick two nodes $v_1$ and $v_2$ from two graphs respectively. Let $K = \lfloor (\frac{1}{2} + \epsilon) \frac{\log 2n}{\log (r-1)} \rfloor$ where $\epsilon$ is a fixed constant, $A_{v_1,G^{(1)}}^{K,spd} = A_{v_2,G^{(2)}}^{K,spd}$ with the probability at most $o(n^{-3/2})$.*

*Proof.* This Lemma can be obtained based on Theorem 6 in [53] with minor corrections. Here we state the proof.

We first introduce the configuration model [54] of $n$-sized $r$-regular graphs. Suppose we have $n$ disjoint sets of items, $W_i, i \in \{1, 2, ..., n\}$, where each set has $r$ items and corresponds to one node in the configuration model. A configuration is a partition of all $nr$ items into $\frac{nr}{2}$ pairs. Denote by $\Omega$ the set of configurations and turn it into a probability space with each configuration the same probability. Turns out among all configurations in $\Omega$, given $r < (2\log n)^{1/2}$, there are about $\exp(-\frac{r^2-1}{4})$ or $\Omega(n^{-1/2})$ portion of them are simple $r$-regular graphs [54]. Then for these configurations, If there is a pair of items with one item from set $W_i$ and another item from set $W_j$, then there is an edge between node $i$ and $j$ in the corresponding $r$-regular graph.

Let $l_0 = \lfloor (\frac{1}{2} + \epsilon) \frac{\log n}{\log (r-1)} \rfloor$, We first look at two nodes randomly selected from the configuration model and consider the following procedure to generate a graph. Let node $i$ and node $j$ be the selected nodes. In the first step, we select all the edges that directly connect to node $i$ and node $j$. Then we have all nodes that are at a distance of 1 to $i$ or $j$. In the second step, we select all the edges that connect to nodes at a distance of 1 to either node $i$ or node $j$. Doing this iteratively for $n - 1$ steps and we end up with the union of components of $i$ and $j$ in a random configuration.

We call an edge is *indispensable* if it is the first edge that ensures a node $w$ is at a certain distance to $\{i, j\}$. Similarly, an edge is *dispensable* if 1) both two ends of the edge are connected to either $i$ or $j$; 2) nodes in both two ends of the edge already have at least one edge. Note that edges with both two ends in the same node are dispensable. As the first $k - 1$ edges selected so far can connect to at most $k + 1$ nodes, the probability that the $k$-th edge selected is dispensable is at most:

$$\frac{(k+1)(r-1)}{(n-k-1)r} \approx \frac{k}{n-k}.$$

Therefore the probability that more than 2 of the first $k_o = \lfloor n^{1/6} \rfloor$ edges are dispensable is at most:

$$\binom{k_o}{3} \left( \frac{k_o}{n-k_o} \right)^3 = o(n^{-2}). \tag{9}$$

The probability that more than $l_1 = \lfloor n^{1/8} \rfloor$ of the first $k_1 = \lfloor n^{6/13} \rfloor$ edges are dispensable is at most

$$\binom{k_1}{l_1+1} \left( \frac{k_1}{n-k_1} \right)^{l_1+1} = o(n^{-2}). \tag{10}$$

The probability that more than $l_2 = \lfloor n^{5/13} \rfloor$ of the first $k_2 = \lfloor n^{2/3} \rfloor$ edges are dispensable is at most

$$\binom{k_2}{l_2+1} \left( \frac{k_2}{n-k_2} \right)^{l_2+1} = o(n^{-2}). \tag{11}$$

Now, let $A$ be the event that at most 2 of the first $k_o$, at most $l_1$ of the first $k_1$, and at most $l_2$ of the first $k_2$ edges be dispensable. Given Equation (9)-(11), we know the probability of $A$ is $1 - o(n^{-2})$.

**Disscussion**: Briefly speaking, event $A$ means that at the first few $k$ steps of generation, the edge will reach almost as many nodes as possible and the number of nodes at distance $\{1, 2, ..., k\}$ will be certainly $r, r(r-1), ..., r(r-1)^{k-1}$ for either $W_i$ and $W_j$. Which means $A_{i,G}^{k,spd} = A_{j,G}^{k,spd}$ with probability close to 1. Nevertheless, it also allows later members of the node configuration to be different.

Now, we consider another way of generating a graph with a configuration model. Similar to the above, suppose we have selected all the edges connecting to at least one node with a distance less

than $k$ from $\{i, j\}$ after the $k$-th step. At the $k + 1$-th step, we first select all the edges with one node at a distance of $k$ from node $i$. Next, we select all the edges with both two ends at a distance of $k$ from node $j$. It is easy to see that after these two procedures, the only edges left for completing the $k + 1$-th step are the edges that have one end at the distance of $k$ from the node $j$ and another end at the distance of $k + 1$ from the node $j$. Suppose there are $t_k$ edges in the nodes at distance $k$ from node $j$ that have not been generated so far and there are $s_k$ nodes that have not generate any edge yet. Then the final procedure of completing the $k + 1$-th step is to connect $t_k$ edges to $s_k$ nodes. This procedure goes on for every $k$ to generate the final graph.

Now let us assume that $A$ holds. It is easily seen that then for $k \leq l_0$, we have:

$$t_k \geq (r - 1)^{k-3} \quad \text{and} \quad s_k \geq n/2, \tag{12}$$

where both two bounds are rather crude. Now, after the first two procedures of $k + 1$-th step, we have all the nodes at the distance of $k + 1$ from node $i$, which means we already determined $Q_{i,G}^{k+1,spd}$. If $|Q_{i,G}^{k+1,spd}| = |Q_{j,G}^{k+1,spd}|$, then the connection of $t_k$ edges must belong to $|Q_{i,G}^{k+1,spd}|$ nodes from totally $s_k$ nodes. The probability of $|Q_{i,G}^{k+1,spd}| = |Q_{j,G}^{k+1,spd}|$ condition on Equation (12) is at most the maximum of the probability that $t_k$ edges connect to $l$ nodes from $s_k$ nodes with degree $r$. This probability is bounded by:

$$max_l \, P(|Q_{j,G}^{k+1,spd}| = l) \leq c_o \frac{s_k^{1/2}}{t_k}, \tag{13}$$

where we assume the $r \geq 3$ and $t_k \leq c s_k^{5/8}$ for some constant $c$ and $c_o$ is also a constant. The proof of this can be found in Lemma 7 of [54]. Given Equation (13), the probability that $A_{i,G}^{l,spd} = A_{j,G}^{l,spd}$ for $l \leq l_0$ is at most

$$1 - P(A) + \prod_{l=h}^{l_0} c_0 \frac{n^{1/2}}{(r-1)^{l-3}},$$

where $h = \lfloor \frac{1}{2} \frac{logn}{log(r-1)} \rfloor + 3$. Since $(r-1)^{l_0} \geq n^{(1+\epsilon)/2}$, the sum above is $o(n^{-2})$. Since there is at least $\Omega(n^{-1/2})$ of all the graphs generated by the configuration model that are simple $r$-regular graphs, there are at most $o(n^{-2}/n^{-1/2}) = o(n^{-3/2})$ of probability that $A_{i,G}^{l_0,spd} = A_{j,G}^{l_0,spd}$.

Next, for any pair of $n$-sized $r$-regular graphs $G^{(1)}$ and $G^{(2)}$, we can combine these two graphs and generate a single regular graph with $2n$ nodes. Denote this combined graph as $G_c$ and $G_c$ has two disconnected components. It is easy to see that the above proof is still valid on $G_c$. This means that: suppose we randomly pick a node $v_1$ from the first component and node $v_2$ from another component. Then, given $3 \leq r < (2log2n)^{1/2}$ and $K = \lfloor (\frac{1}{2} + \epsilon) \frac{\log 2n}{\log (r-1)} \rfloor$, we have $A_{v_1,G_c}^{K,spd} = A_{v_2,G_c}^{K,spd}$ with probability of $o(n^{-3/2})$. As node $v_1$ and $v_2$ are in two disconnected components $G^{(1)}$ and $G^{(2)}$. Therefore, it is easy to see $A_{v_1,G_c}^{K,spd} = A_{v_1,G^{(1)}}^{K,spd}$ and $A_{v_2,G_c}^{K,spd} = A_{v_1,G^{(2)}}^{K,spd}$, which completes the proof.

$\square$

Theorem 1 is easy to prove with the aid of Lemma 1. Basically, we consider a node $v_1$ in graph $G^{(1)}$ and compare the $A_{v_1,G^{(1)}}^{K,spd}$ with $A_{v_2,G^{(2)}}^{K,spd}$ for all nodes $v_2 \in V^{(2)}$. The probability that $A_{v_2,G^{(2)}}^{K,spd} \neq A_{v_1,G^{(1)}}^{K,spd}$ for all possible $v_2$ is $1 - o(n^{-3/2}n) = 1 - o(n^{-1/2})$. Therefore, with an injective readout function, we can guarantee that 1 layer $K$-hop message passing GNN can generate different embedding for two graphs.

### C.2 Simulation experiments to verify Theorem 1

In this section, we conduct simulation experiments to verify both Lemma 1 and Theorem 1. The results are shown in Figure 4. We randomly generate 100 $n$-sized 3 regular graphs with $n$ ranging from 20 to 1280. Then, we apply a 1-layer untrained K-GIN model on these graphs (1 as node

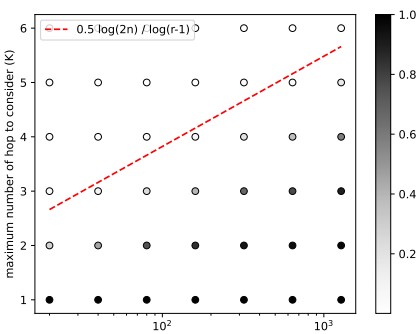 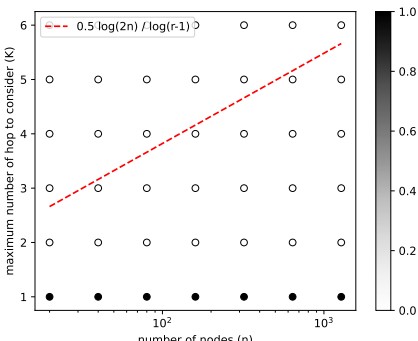

Figure 4: Simulation results. The left side is the node-level result to verify Lemma 1. The right side is the graph-level result to verify Theorem 1.

feature) with $K$ range from 1 to 6. On the left side, we compare the final node representations for all nodes output by K-GIN, If the difference between two node representations $||h_v - h_u||_2$ is less than machine accuracy ($1e - 10$), they are regarded as indistinguishable. The colors of the scatter plot indicate the portion of two nodes that are not distinguishable by K-GIN. The darker, the more indistinguishable node pairs. We can see that the result matches almost perfectly with Lemma 1, where $K$ is larger than $\frac{1}{2}\frac{\log 2n}{\log (r-1)}$, almost all nodes are distinguishable by K-GIN. On the right side, we compare the final graph representation output by K-GIN. We can see even with $K = 2$, almost all graphs are distinguishable. This is because as long as there exists one single node from one graph that has a different representation from all nodes in another graph, 1 layer K-hop message passing with an injective readout function can distinguish two graphs.

## D   General $K$-hop color refinement and discussion on existing $K$-hop message passing GNNs

In this section, we introduce a general $K$-hop color refinement algorithm and use this algorithm to characterize the expressive power of existing $K$-hop methods further.

### D.1   General $K$-hop color refinement algorithm

It is well-known that 1-WL test updates the label of each node in the graph by color refinement algorithm, which iteratively aggregates the label of its neighbors. Here, we extend it and define a more general color refinement algorithm. First, we denote $[n] = \{0, 1, ..., n\}$ and introduce *refinement configuration*.

**Definition 6.** *Given $L$ and $K$, the refinement configuration $\mathbf{C}^{L,K} = (C^{0,K}, C^{1,K}, ..., C^{L,K})$, where $C^{l,K} = (C_0^{l,K}, C_1^{l,K}, ..., C_K^{l,K})$ and $C_k^{l,K} \subseteq [l]$ for any $l \in [L]$ and $k \in [K]$.*

Briefly speaking, refinement configuration defines how the color refinement algorithm aggregates information from neighbors of each hops at each iteration $l$ given the maximum iteration of $L$ and the maximum number of hop $K$. Given the refinement configuration $\mathbf{C}^{L,K}$, we define general color refinement algorithm $\mathbf{CR}(\mathbf{C}^{L,K})$:

$$R_{v,G}^0 = \text{LABEL}(v),$$

$$R_{v,G}^{l+1} = \text{HASH}((\{\!\{\{\!\{R_{u,G}^s | u \in Q_{v,G}^{0,t}\}\!\} | s \in C_0^{l,K}\}\!\}, ..., \{\!\{\{\!\{R_{u,G}^s | u \in Q_{v,G}^{K,t}\}\!\} | s \in C_K^{l,K}\}\!\})), \tag{14}$$

where LABEL function assign the initial color to node. Equation (14) define the general color refinement algorithm. Then, given different refinement configurations, we can end up with different procedures for the algorithm. Specifically, we define the following refinement configurations:

**Definition 7.** *The 1-WL refinement configuration is defined as $\mathbf{C}_{1\text{-}WL}^{L,K} = (C_{1\text{-}WL}^{0,K}, C_{1\text{-}WL}^{1,K}, ..., C_{1\text{-}WL}^{L,K})$, where $C_{1\text{-}WL,k}^{l,K} = \{l\}$ for $k = 0, 1$, and $C_{1\text{-}WL,k}^{l,K} = \emptyset$ for others.*

**Definition 8.** *The $K$-hop refinement configuration is defined as $\mathbf{C}_{K\text{-}hop}^{L,K} = (C_{K\text{-}hop}^{0,K}, C_{K\text{-}hop}^{1,K}, ..., C_{K\text{-}hop}^{L,K})$, where $C_{K\text{-}hop,k}^{l,K} = \{l\}$ for all $k \in [K]$.*

**Definition 9.** *The GINE+ refinement configuration is defined as* $\mathbf{C}_{GINE+}^{L,K} = (C_{GINE+}^{0,K}, C_{GINE+}^{1,K}, ..., C_{GINE+}^{L,K})$, *where* $C_{GINE+,k}^{l,K} = \{l\}$ *for* $k = 0$, *and* $C_{GINE+,k}^{l,K} = \{l - k + 1\}$ *for* $k = 1, 2, ..., l + 1$, $C_{GINE+,k}^{l,K} = \emptyset$ *if* $k > l + 1$.

It is easy to see that $\mathbf{CR}(\mathbf{C}_{1\text{-WL}}^{L,K})$ is exactly the same as the color refinement algorithm in 1-WL test. Next, we can analyze the expressive power of the general color refinement algorithm given different refinement configurations. We say $\mathbf{C}_1^{L,K} \succeq \mathbf{C}_2^{L,K}$ if $\mathbf{CR}(\mathbf{C}_1^{L,K})$ is at least equally powerful as $\mathbf{CR}(\mathbf{C}_2^{L,K})$ in terms of expressive power. Then, we have the following properties for the general refinement algorithm.

**Property 1.** $\mathbf{C}^{L+1,K} \succeq \mathbf{C}^{L,K}$.

**Property 2.** *If* $C_{1,k}^{l,K} \subseteq C_{2,k}^{l,K}$ *for any* $k \in [K]$ *and* $l \in [L]$, *then* $\mathbf{C}_1^{L,K} \succeq \mathbf{C}_2^{L,K}$.

These two properties are easy to validate, as given the injective HASH function, an algorithm with more information is always at least equally powerful as an algorithm with less information. Moreover, we have the following proposition.

**Proposition 4.** *If for any* $l \in [L]$ *and any* $k \in [K]$, $\mathbf{C}_1^{l,K}$ *and* $\mathbf{C}_2^{l,K}$ *satisfy* $i \in C_{1,k}^{l,K}$ *and* $j \in C_{2,k}^{l,K}$ *if and only if* $i \geq j$, *then* $\mathbf{C}_1^{L,K} \succeq \mathbf{C}_2^{L,K}$.

*Proof.* Let $R_{v,G}^{l,i}$ denote the color refinement result after iteration $l$ for node $v$ in graph $G = (V, E)$ using refinement configuration $\mathbf{C}_i^{L,K}$.

1. At iteration 1, if a node aggregates its neighbor's label from some hop, it can only aggregate the initial label of nodes, or namely $R_{v,G}^{0,i}$. Then, if $C_{1,k}^{1,K} = \{\!\{0\}\!\}$, $C_{2,k}^{1,K}$ can be either $\{\!\{0\}\!\}$ or $\emptyset$. If $C_{1,k}^{1,K} = \emptyset$, then $C_{2,k}^{1,K} = \emptyset$ It is trivial to see that if $R_{v_1,G}^{1,1} = R_{v_2,G}^{1,1}$ for any pair of nodes $v_1, v_2 \in G$, then $R_{v_1,G}^{1,2} = R_{v_2,G}^{1,2}$, which means $\mathbf{C}_1^{0,K} \succeq \mathbf{C}_2^{0,K}$ holds.

2. At iteration $l$ Assume $\mathbf{C}_1^{l-1,K} \succeq \mathbf{C}_2^{l-1,K}$ holds.

3. At iteration $l + 1$, the condition in the proposition means color refinement algorithm with $C_2^{l,K}$ can only aggregate results from earlier iteration than $C_1^{l,K}$ at any hops. Meanwhile, as $\mathbf{C}_1^{l-1,K} \succeq \mathbf{C}_2^{l-1,K}$ holds, as long as $R_{v_1,G}^{l,1} = R_{v_2,G}^{l,1}$, we have $R_{v_1,G}^{t,i} = R_{v_2,G}^{t,i}$ holds for any $t \in [l]$ and $i = 1, 2$ given the injectiveness of HASH function. This means that if $R_{v_1,G}^{l+1,1} = R_{v_2,G}^{l+1,1}$, then $R_{v_1,G}^{l+1,2} = R_{v_2,G}^{l+1,2}$. Therefore, $\mathbf{C}_1^{l,K} \succeq \mathbf{C}_2^{l,K}$ also holds. this completes the proof.

$\square$

Given two properties and Proposition 4, we have the following results.

**Theorem 3.** $\mathbf{C}_{K\text{-hop}}^{L,K} \succeq \mathbf{C}_{GINE+}^{L,K} \succeq \mathbf{C}_{1\text{-WL}}^{L,K} \succeq \mathbf{C}_{GINE+}^{0,K} \succeq \mathbf{C}_{1\text{-WL}}^{0,K}$.

*Proof.* Using the Property 1 and Property 2 of general color refinement algorithm, it is easy to prove that $\mathbf{C}_{K\text{-hop}}^{L,K} \succeq \mathbf{C}_{1\text{-WL}}^{L,K} \succeq \mathbf{C}_{GINE+}^{0,K} \succeq \mathbf{C}_{1\text{-WL}}^{0,K}$ and $\mathbf{C}_{GINE+}^{L,K} \succeq \mathbf{C}_{1\text{-WL}}^{L,K}$. The only thing left is the comparison between $\mathbf{C}_{K\text{-hop}}^{L,K}$ and $\mathbf{C}_{GINE+}^{L,K}$. For any $l \in [L]$, $C_{K\text{-hop},k}^{l,K} = \{\!\{l\}\!\}$ for all $k \in [K]$. Instead, $C_{GINE+,k}^{l,K} = \{\!\{l - k + 1\}\!\}$ for $k = 1, 2, ..., l + 1$ and $C_{GINE+,0}^{l,K} = \{\!\{l\}\!\}$. Then it is easy to see that $C_{K\text{-hop},k}^{l,K}$ and $C_{GINE+,k}^{l,K}$ satisfy the condition of Proposition 4 and thus $\mathbf{C}_{K\text{-hop}}^{L,K} \succeq \mathbf{C}_{GINE+}^{L,K}$ holds. $\square$

Theorem 3 provide a general comparison between different color refinement configurations. Based on Theorem 3, we can actually extend the Proposition 1 in the main paper as

**Corollary 1.** *$L$ layer $K$-hop message passing GNNs defined in Equation (3) with the shortest path distance kernel is at least equally powerful as $L$ layer GINE+ [16]. $L$ layer GINE+ is strictly more powerful than $L$ layer 1-hop message passing GNNs.*

The above Corollary is trivial to prove as corresponding models with injective message and update functions have at most the same expressive power as general color refinement algorithm with corresponding refinement configurations and permutation invariant readout function. However, one remaining question given the general color refinement algorithm is the comparison of the expressive power between $K$ layer 1-hop GNNs and 1 layer $K$-hop GNNs. Here we show that:

**Proposition 5.** *Assume we use the shortest path distance kernel for $K$-hop message passing GNNs. There exists pair of graphs that can be distinguished by 1 layer $K$-hop message passing GNNs but not $K$ layer 1-hop message passing GNNs and vice versa.*

To prove Proposition 5, we provide two examples. The first example is exactly example 2 in Figure 1. we know that a 1-hop GNN with 2 layers cannot distinguish two graphs as they are all regular graphs. Instead, a 1-layer 2-hop GNN is able to achieve that. From another direction, we show two graphs in Figure 5. We know that 1 layer 2-hop message passing GNN is equivalent to inject node configuration up to 2-hop into nodes. As we show in Figure 5, two graphs have the same node configuration set, which means an injective readout function will produce the same representation. Therefore, two graphs cannot be distinguished by 1 layer 2-hop message passing GNNs. Instead, it is easy to validate that these two graphs can be distinguished by 2 layer 1-hop message passing GNNs.

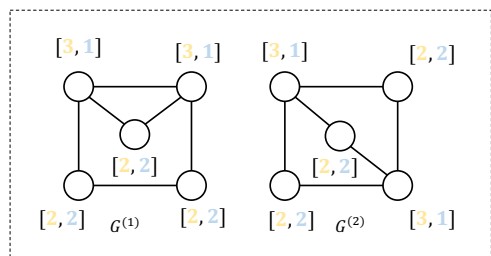

Figure 5: A pair of non-isomorphic graphs that can be distinguished by 2-layer 1-hop GNNs but not 1-layer 2-hop GNNs.

### D.2 Discussion on existing $K$-hop models

**GINE+** [16]: GINE+ tries to increase the representation power of graph convolution by increasing the kernel size of convolution. However, at $l$-th layer of GINE+, it only aggregates information from the neighbors of hop $1, 2, ..., l$ after $l - 1, l - 2, ..., 0$ layer, which means that after $L$ layer of convolution, the GINE+ still has a receptive field of size $L$. As we discussed in the previous section, $L$ layer $K$-hop message passing GNNs with the shortest path distance kernel is at least equally powerful as $L$ layer GINE+.

**Graphormer** [17]: Graphormer introduce a new way to apply transformer architecture [55] on graph data. In each layer of Graphormer, the shortest path distance is used as spatial encoding to adjust the attention score between each node pair. Although the Graphormer does not apply the $K$-hop message passing directly, the attention mechanism (each node can see all the nodes) with the shortest path distance feature implicitly encodes a rooted subtree similar to the $K$-hop message passing with the shortest path distance kernel. To see it clearly, suppose we have $K$-hop message passing with $K = \infty$ and graphs only have one connected component. It will aggregate all the nodes at each layer, which is similar to Graphormer. Meanwhile, the injective message and update function implicitly encode the shortest path distance to each node in aggregation. Then it is trivial to see that Graphormer is actually a special case of $K$-hop message passing with the shortest path distance kernel.

**Spectral GNNs**: spectral-based GNNs serve as an important type of graph neural network and gain lots of interest in recent years. Here we only consider one layer as spectral-based GNNs usually only use 1 layer. the general spectral GNNs can be written as:

$$Z = \phi \left( \sum_{k=0}^{K} \alpha_k \rho(\hat{L}^k) \varphi(x) \right), \tag{15}$$

where $\phi$ and $\varphi$ are typically multi-layer perceptrons (MLPs), $\hat{L}$ is normalized Laplacian matrix, $\alpha_k$ is weight for each spectral basis, and $\rho$ is an element-wise function of matrix. In normal spectral GNNs, $\rho$ is always an identity mapping function. We can see spectral GNNs have a close relationship to $K$-hop message passing with graph diffusion kernel as $\hat{L}^k$ actually compute the $Q_{v,G}^{k,gd}$ for each node by only keeping element in the matrix that is larger than 0. As $K$ in Equation (15) can be greater than 1, it looks like spectral GNNs fit Proposition 1 as well. However, according to Proposition 4.3 in [21],

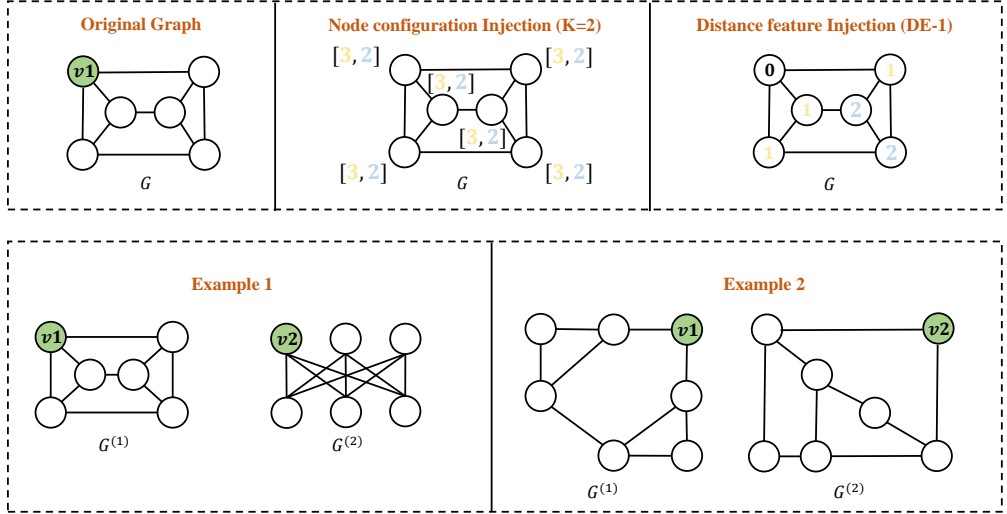

Figure 6: The upper part: graph with node configuration as the injected label and DE-1 as the injected label on the center node. The bottom part: two pairs of non-isomorphic graphs where the node pair in example 1 can be distinguished by DE-1 and the node pair in example 2 can be distinguished by $K$-hop message passing.

all such spectral-based methods have expressive power no more than the 1-WL test. This seems like a discrepancy. The reason lies in the $\rho$ function in Equation (15). $K$-hop message passing with graph diffusion kernel can be regarded as using a non-linear $\rho$ function shown in the following:

$$\rho(x) = \begin{cases} 1 & x > 0 \\ 0 & others \end{cases} \tag{16}$$

This non-linear function is the key difference between the normal spectral GNNs and $K$-hop message passing and it endows normal spectral GNNs with extra expressive power than the 1-WL test.

**MixHop [12], GPR-GNN [15], and MAGNA [14]**: All three methods extend the scope of 1-hop message passing by considering multiple graph diffusion step simultaneously. However, all these methods use normal spectral GNNs as the base encoder and thus they obey Proposition 4.3 in [21] instead of Proposition 1. It can be seen as a "weak" version of $K$-hop message passing with graph diffusion kernel.

## E   Comparison between $K$-hop message passing and Distance Encoding

In this section, we further discuss the connection and difference between $K$-hop message passing and Distance Encoding [22]. Here we assume that the kernel of $K$-hop message passing is the shortest path distance kernel and Distance Encoding with the shortest path distance as distance feature.

To simplify the discussion, suppose there are two graphs $G^{(1)} = (V^{(1)}, E^{(1)})$ and $G^{(2)} = (V^{(2)}, E^{(2)})$. We pick two nodes $v_1$ and $v_2$ from each graph respectively and learn the representation for these two nodes. First, let us consider what 1 layer $K$-hop message passing and DE-1 without message passing. As we stated in the main paper, 1 layer of $K$-hop message passing actually injects each node with the label of node configuration. Instead, DE-1 injects each node with the distance to the center node ($v_1$ and $v_2$ here). Then we can see there is a clear difference between the two methods even if they all implicitly or explicitly use the distance feature as shown in the upper part of Figure 6. Next, it is easy to see that applying $L + 1$ layer $K$-hop message passing is equivalent to applying $L$ layer $K$-hop message passing on a graph with node configuration as the initial label. Applying $L$ layer DE-1 is equivalent to applying $L$ layer 1-hop message passing on a graph with the distance to the center node as the initial label. In the following, we show that the two methods cannot cover each other in terms of distinguishing different nodes:

**Proposition 6.** *For two non-isomorphic graphs $G^{(1)} = (V^{(1)}, E^{(1)})$ and $G^{(2)} = (V^{(2)}, E^{(2)})$, we pick two nodes $v1$ and $v_2$ from each graph respectively. Then there exist pairs of graphs that nodes $v_1$ and $v_2$ can be distinguished by $K$-hop message passing with the shortest path distance kernel but not DE-1, and vice versa.*

To prove the Proposition 6, we provide two examples in the lower part of Figure 6. Example 1 in the figure is exactly the same pair of regular graphs in example 1 of Figure 1. As discussed before, $K$-hop message passing with the shortest path distance kernel cannot distinguish node $v_1$ and $v_2$. However, after injecting the distance feature to the center node in two graphs and performing 2 layers of message passing, DE-1 is able to assign different representations for two nodes. In example 2, DE-1 cannot distinguish nodes $v_1$ and $v_2$ in the two graphs, even if they are not regular graphs. Instead, using $K$-hop message passing with only $K = 2$, the two nodes will get different representations. We omit the detailed procedure here as it is easy to validate. Using these two examples, we have shown that $K$-hop message passing and DE-1 are not equivalent to each other even if they all use the shortest path distance feature. The root reason is that although DE-1 uses the distance feature, it still aggregates information only from 1-hop neighbors in each iteration, while $K$-hop message passing directly aggregates information from all $K$-hop neighbors. That is, their ways of using distance information are different. However, we also notice that in example 2, if we label the graph with DE-1 on another pair of nodes, two nodes can be distinguished, which means DE-1 is able to distinguish these two graphs. **However, to achieve that, DE-1 need to label the graph $n$ times and run the message passing on all $n$ labeled graphs. Instead, $K$-hop message passing only needs to run the message passing once, which is both space and time efficient.**

Besides DE-1, Li et al. [22] also proposed the DEA-GNN which is at least no less powerful than DE-1. The DEA-GNN extends DE-1 by simultaneously aggregating all other nodes in the graph but the message is encoded with the distance to the center node. This can be seen as exactly performing $K$-hop message passing. In other words, DEA-GNN is the combination of $K$-hop message passing and DE-1. Therefore it is easy to see:

**Proposition 7.** *The DEA-GNN [22] is at least equally powerful as $K$-hop message passing with the shortest path distance kernel.*

## F   Proof of Theorem 2

In this section, we prove Theorem 2. We first restate Theorem 2: The expressive power of a proper $K$-hop message passing GNN of any kernel is bounded by 3-WL test. Our proof is inspired by the recent results in SUN [23], which bound all subgraph-based GNN with the 3-WL test by proving that all such methods can be implemented by 3-IGN. Here we prove that $K$-hop message passing can also be implemented by 3-IGN. We will not discuss the detail of 3-IGN and all its operations. Instead, we directly follow all the definitions and notations and refer readers to Appendix B of [23] for more details.

$K$**-hop neighbor extraction**: To implement the $K$-hop message passing, we first implement the extraction of $K$-hop neighbors. The key insight is that we can use the $l$-th channel in $X_{iij}$ to store whether node $j$ is the neighbor of node $i$ at $l$-th hop, which is similar to the extraction of ego-network. Here we suppose $d \geq K$. Same as all node selection policies, we first lift the adjacency $A$ to a three-way tensor $\mathcal{Y} \in \mathbb{R}^{n^3 \times d}$ using broadcasting operations:

$$X_{iii}^{(0)} = \boldsymbol{\beta}_{i,i,i} A_{ii}. \tag{17}$$

$$X_{jii}^{(0)} = \boldsymbol{\beta}_{*,i,i} A_{ii}. \tag{18}$$

$$X_{iij}^{(0)} = \boldsymbol{\beta}_{i,i,j} A_{ij}. \tag{19}$$

$$X_{iji}^{(0)} = \boldsymbol{\beta}_{i,j,i} A_{ij}. \tag{20}$$

$$X_{kij}^{(0)} = \boldsymbol{\beta}_{*,i,j} A_{ij}. \tag{21}$$

Now, $X_{iij}^{(0),1}$ store the 1-hop neighbors of node $i$. Next, $l$-th hop neighbor of node $i$ is computed and stored in $X_{iij}^{;l}$. To get neighbor of $l$-th hop for $l = 2, 3, ..., K$, we first copy it into $d + 1$ channels:

$$X_{ijj}^{(1)} = \boldsymbol{\kappa}_{:d/(d+1)}^{:d} X_{ijj}^{(0)} + \boldsymbol{\kappa}_{d+1:d+1}^{l:l} \boldsymbol{\beta}_{i,j,j} X_{iij}^{(0)}. \tag{22}$$

The $d + 1$-th channel is used to compute high-order neighbors. Next, we extract all $K$ hop neighbors by iteratively following steps $K - 1$ times and describe the generic $l$-th step. We first broadcast the current reachability pattern into $X_{ijk}$, writing it into the $l$-th channel:

$$X_{ijk}^{(l,1)} = X_{ijk}^{(l-1)} + \boldsymbol{\kappa}_{l:l}^{d+1:d+1} \boldsymbol{\beta}_{i,*,j} X_{ijj}^{(l-1)}. \tag{23}$$

Then, a logical AND is performed to get the new reachability. Then write back the results into the $l$-th channel:

$$X_{ijk}^{(l,2)} = \boldsymbol{\varphi}_{l:l/(d+1)}^{(\wedge)1,l} X_{ijk}^{(l,1)}. \tag{24}$$

Next, we get all nodes that can be reached within $l$ hops by performing pooling, clipping, and copying back to $d + 1$ channel:

$$X_{ijj}^{(l,3)} = \boldsymbol{\kappa}_{:d/(d+1)}^{:d} X_{ijj}^{(l,2)} + \boldsymbol{\kappa}_{d+1:}^{l:l} \boldsymbol{\beta}_{i,j,j} \boldsymbol{\pi}_{i,j} X_{ijk}^{(l,2)}. \tag{25}$$

$$X_{ijj}^{(l,4)} = \left[ \begin{array}{cc} \boldsymbol{\kappa}_{:d/(d+1)}^{:d} & \boldsymbol{\varphi}_{d+1:d+1}^{(\downarrow)d+1:d+1} \end{array} \right] X_{ijj}^{(l,3)}. \tag{26}$$

Now $X_{ijj}^{(l,4),d+1}$ save all the nodes that can be reached at $l$-th hop. Finally, we extract the $l$-th hop neighbor and copy it into $l$-th channel. For graph diffusion kernel, the current result is itself result for graph diffusion kernel, which means we only need to copy it:

$$X_{iij}^{(l)} = X_{iij}^{(l,4)} + \boldsymbol{\kappa}_{l:l}^{d+1:d+1} \boldsymbol{\beta}_{i,i,j} X_{ijj}^{(l,4)}. \tag{27}$$

For the shortest path distance kernel, we need to nullify all the nodes that already existed in the previous hops. To achieve this, we need to first compute if a node exists in the previous hops and then nullify it:

$$X_{iij}^{(l,5)} = X_{iij}^{(l,4)} + \boldsymbol{\kappa}_{l:l}^{d+1:d+1} \boldsymbol{\beta}_{i,i,j} X_{ijj}^{(l,4)}. \tag{28}$$

$$X_{iij}^{(l,6)} = \sum_{i=1}^{l-1} \boldsymbol{\kappa}_{d+1:d+1}^{i:i} X_{iij}^{(l,5)}. \tag{29}$$

$$X_{iij}^{(l)} = X_{iij}^{(l,4)} + \boldsymbol{\varphi}_{l:l}^{(\wedge)l,d+1} \boldsymbol{\varphi}_{d+1}^{(!)d+1} X_{iij}^{(l,6)}. \tag{30}$$

Where $\boldsymbol{\varphi}_b^{(!)a}$ is logical not function that output 0 if input is not 0 and vice versa for channel $a$ and write result into channel $b$. Here we omit the detailed implementation as it is easy to implement using ReLu function. Finally, we bring all other orbits tensors to the original dimensions:

$$X_{iii} = \boldsymbol{\kappa}_{:d}^{:d} X_{iii}^{(l)}. \tag{31}$$

$$X_{ijj} = \boldsymbol{\kappa}_{:d}^{:d} X_{ijj}^{(l)}. \tag{32}$$

$$X_{iij} = \boldsymbol{\kappa}_{:d}^{:d} X_{iij}^{(l)}. \tag{33}$$

$$X_{iji} = \boldsymbol{\kappa}_{:d}^{:d} X_{iji}^{(l)}. \tag{34}$$

$$X_{ijk} = \boldsymbol{\kappa}_{:1/(d)}^{:1} X_{ijk}^{(l)}. \tag{35}$$

Now, we have successfully implemented $K$-hop neighbor extraction algorithm using 3-IGN.

$K$-**hop message passing**: To implement the message and update function for each layer, we use the same base encoder as [11]. Other types of base encoders and combine functions can be implemented

in a similar way. We follow the same procedure of implementing the base encoder of DSS-GNN as stated in [23]. Note here for each hop the procedure is similar therefore we state the generic $l$-th hop. The key insight is that, in $K$-hop message passing, we are actually not working on each subgraph but the original graph, which means all the operations can be implemented in the orbit tensor $X_{iii}$ and $X_{iij}$.

*Message broadcasting*: This procedure is similar to the base encoder of DSS-GNN but here we only need to broadcast $X_{iij}$. However, since we need to perform message passing for $K$ times, we need to broadcast it $K$ times:

$$X_{iij}^{(t,1)} = \boldsymbol{\kappa}_{:d/((K+1)d)}^{:d} X_{iij}^{(t)} + \sum_{i=1}^{K} \boldsymbol{\kappa}_{id+1:(i+1)d/(K+1)d}^{:} \boldsymbol{\beta}_{iij} X_{ijj}^{(t)}. \tag{36}$$

*Message sparsification and aggregation*: Similar to DSS-GNN, here we need to nullify the message from nodes that are not $l$-th hop neighbors. Here we define the following function:

$$f_{iij}^{\odot} \left( X_{aab}^{(t,1)} \right)_l = \begin{cases} \mathbf{0}_d & \text{if } X_{aab}^{(t,1),l} = 0, \\ X_{aab}^{(t,1),l(d+1):(l+1)d} & \text{otherwise.} \end{cases} \tag{37}$$

The function is used to nullify the message for $l$-th hop. It is easy to validate the existence of such function following the same procedure in [23] therefore we omit the detail. Next, the message sparsification can be implemented by:

$$X_{iij}^{(t,3)} = \begin{bmatrix} \boldsymbol{\kappa}_{:d}^{:d} & \boldsymbol{\varphi}_1^{(\odot_{ijj}):} & \dots & \boldsymbol{\varphi}_K^{(\odot_{ijj}):} \end{bmatrix} X_{iij}^{(t,2)}. \tag{38}$$

Then, the message function for $K$-hop message passing is:

$$X_{iii}^{(t,4)} = \boldsymbol{\kappa}_{:d}^{:d} X_{iii}^{(t,3)} + \boldsymbol{\kappa}_{d+1:}^{d+1:} \boldsymbol{\beta}_{iii} \boldsymbol{\pi}_i X_{ijj}^{(t,3)}. \tag{39}$$

*Update* Then, the update function is implemented using linear transformation. In $K$-hop message passing, each hop needs an independent parameter set. Here in order to operate on constructed tensor, we define $W_t^l = \begin{bmatrix} W_{t,1}^l || W_{t,2}^l || ... || W_{t,K}^l \end{bmatrix}$, where $W_{t,i}^l = \mathbf{0}_d$ if $i \neq l$. Then, the update function is:

$$X_{iii}^{(t,5)} = \sum_{l=1}^{K} \sigma \left( W_t^l X_{iii}^{(t,4)} \right). \tag{40}$$

*Combine*: Finally, we implement the sum combine function. The combine function is can be implemented by a simple MLP:

$$X_{iii}^{(t+1)} = \boldsymbol{\varphi}_d^{\boldsymbol{f}} (X_{iii}^{(t,4)}). \tag{41}$$

$$X_{jii}^{(t+1)} = \boldsymbol{\beta}_{jii} X_{iii}^{t+1}. \tag{42}$$

Now, we successfully implement both $K$-hop neighbor extraction and $K$-hop message passing layer. Note that other parts like the readout function can be easily implemented and we omit the detail. This means the expressive power of $K$-hop message passing with either graph diffusion kernel or shortest path distance kernel is bounded by 3-IGN. Based on the Theorem 2 in [56], we conclude the Theorem 2. Further, it is intuitive that all methods that use the node pair distance feature are also bounded by 3-WL as this feature can be computed by 3-IGN. We leave the formal proof in our future work.

# G   Proof and discussion of Proposition 2

Proposition 2 seems intuitive and easy to prove. However, it also unclear how powerful are KP-GNN. So here we still give a more detailed discussion on it. we first give the definition of *distance regular graph*.

**Definition 10.** *A distance regular graph is a regular graph such that for any two nodes $v$ and $u$, the number of nodes with the distance of $i$ from $v$ and distance of $j$ from $u$ depends only on $i$, $j$ and the distance between $v$ and $u$.*

Furthermore, we only consider the connected distance regular graphs with no multi-edge or self-loop. Such graphs can be characterized by *intersection array*.

**Definition 11.** *The intersection array of a distance regular graph with diameter $d$ is an array of integers $(b_0, b_1, ..., b_{d-1}; c_1, c_2, ..., c_d)$, where for all $1 \leq j \leq d$, $b_j$ gives the number of neighbors of $u$ with distance $j + 1$ from $v$ and $c_j$ gives the number of neighbors of $u$ with distance $j - 1$ from $v$ for any pair of $(v, u)$ in graph with distance $j$.*

Given the definition of distance regular graph and intersection array, we can propose the first lemma.

**Lemma 2.** *Given a distance regular graph $G$ with intersection array $(b_0, b_1, ..., b_{d-1}; c_1, c_2, ..., c_d)$. Pick a node $v$ from $G$, the peripheral subgraph $G_{v,G}^{j,spd}$ is a $n$-sized $r$-regular graph with $n = |Q_{v,G}^{j,spd}|$ and $r = b_0 - b_j - c_j$*

*Proof.* Given a distance regular graph $G$ with intersection array $(b_0, b_1, ..., b_{d-1}; c_1, c_2, ..., c_d)$, from the definition of intersection array, for node $v$ in $G$, $Q_{v,G}^{j,spd}$ is the node set that have distance $j$ from node $v$. Then, $b_j$ is the number of neighbors for each node in $Q_{v,G}^{j,spd}$ that have distance $j + 1$ to node $v$. It is easy to see that these neighbors must belong to $Q_{v,G}^{j+1,spd}$, which means that $b_j$ is also the number of edge for a node in $Q_{v,G}^{j,spd}$ that connect to nodes in $Q_{v,G}^{j+1,spd}$. Similarly, $c_j$ is the number of edge for a node in $Q_{v,G}^{j,spd}$ that connect to nodes in $Q_{v,G}^{j-1,spd}$. For node $u \in Q_{v,G}^{j,spd}$, we know that the edges of $u$ must connect to either $Q_{v,G}^{j+1,spd}$, $Q_{v,G}^{j,spd}$, or $Q_{v,G}^{j-1,spd}$. Since the degree of node $u$ is $b_0$, then the number of edge that connect node $u$ to nodes in $Q_{v,G}^{j,spd}$ is $b_0 - b_j - c_j$. The above statement holds for each $u \in Q_{v,G}^{j,spd}$, which means all nodes $u \in Q_{v,G}^{j,spd}$ have same node degree. Meanwhile, the node set of peripheral subgraph $G_{v,G}^{j,spd}$ is exactly $Q_{v,G}^{j,spd}$. Combine two statements, we can conclude that $G_{v,G}^{j,spd}$ is a $n$-sized $r$-regular graph with $n = |Q_{v,G}^{j,spd}|$ and $r = b_0 - b_j - c_j$.   □

Given the Lemma 2, we know that the peripheral subgraph of a node in any distance regular graph is itself a regular graph. Next, given two non-isomorphic distance regular graphs $G^{(1)}$ and $G^{(2)}$ with the same intersection array, there are total $d$ pairs of regular peripheral subgraphs for any pair of nodes $v_1$ and $v_2$. If the KP-GNN can distinguish two regular graphs at some hop $j \leq d$, then the KP-GNN can distinguish $v_1$ and $v_2$ in graph $G^{(1)}$ and $G^{(2)}$. Meanwhile, it is easy to see that each node in a distance regular graph has the same local structure. Therefore, as long as $v_1$ can be distinguished from $v_2$, KP-GNN can distinguish two graphs.

Now given the implementation of KP-GNN in Equation 5, if either the peripheral set $E(Q_{v_1,G^{(1)}}^{k,t})$ is different from $E(Q_{v_2,G^{(2)}}^{k,t})$ or peripheral configuration $C_{j,G^{(1)}}^{k'}$ is different from $C_{j,G^{(2)}}^{k'}$. However, as peripheral subgraph is itself regular graphs, $E(Q_{v_1,G^{(1)}}^{k,t})$ must equal to $E(Q_{v_2,G^{(2)}}^{k,t})$. Therefore, KP-GNN can only distinguish two peripheral subgraph by $C_{j,G^{(1)}}^{k'}$ and $C_{j,G^{(2)}}^{k'}$.

It looks like we can use the result from Theorem 1 to prove that Equation 5 can distinguish almost all distance regular graphs. However, although the peripheral subgraphs of $v_1$ and $v_2$ are regular graphs with the same size and $r$, they are not randomly generated by the configuration model and do not satisfy Theorem 1. We leave the quantitative expressive power analysis of KP-GNN in our future works.

# H Time, space complexity and limitation of $K$-hop message passing and KP-GNN

## H.1 Time and space complexity

In this section, we analyze the time and space complexity. To simplify the analysis, we first consider the shortest path distance kernel, as the graph diffusion kernel has both time and space complexity no less than the shortest path distance kernel. Denote graph $G$ with $n$ node and $m$ edges.

**Space complexity**: For both $K$-hop message passing and KP-GNN, as we only need to maintain one representation for each node, the space complexity is $O(n)$ like in vanilla 1-hop message passing.

**Time complexity**: First, we analyze the time complexity of $K$-hop message passing GNNs. For each node, suppose in the worst case we extract neighbors from all nodes from all hops, the number of neighbors we need to aggregate from all hops is $n$ (all nodes in the graph). Then, $n$ nodes need at most $O(n^2)$ time complexity. Next, we analyze the time complexity of KP-GNN. The additional time complexity comes from counting the peripheral edges and $k'$-configuration. Since the counting can be done in a preprocessing step and reused at each message passing iteration, it will be amortized to zero finally. So the practical time complexity is still $O(n^2)$.

## H.2 Discussion on the complexity

From the above analysis, we know that $K$-hop GNNs and KP-GNN only need $O(n)$ space complexity, which is equal to vanilla message passing GNNs and much less than subgraph-based GNNs which require $O(n^2)$ space (due to maintaining a different representation for each node when appearing in different subgraphs). Thus, KP-GNN has a much better space complexity than subgraph-based GNNs, PPGN [24] (also $O(n^2)$) and 3-WL-GNNs [25] ($O(n^3)$).

The worst-case time complexity of $K$-hop GNNs and KP-GNN is much higher than that of normal GNNs due to aggregating information from more than 1-hop nodes in each iteration. However, we also note that the larger receptive field could reduce the number of message passing iterations because 1-layer of $K$-hop message passing can cover the receptive field of $K$-layer 1-hop message passing. Furthermore, $K$-hop GNN and KP-GNN have better time complexity $O(n^2)$ than that of subgraph-based GNNs (which require $O(nm)$ for doing 1-hop message passing $O(m)$ in all $n$ subgraphs). For sparse graphs, we can already save a factor of $d_{\text{avg}} = m/n$ complexity. For dense graphs, the worst-case time complexity of subgraph-based GNNs becomes $O(n \cdot n^2) = O(n^3)$, and our time complexity advantage becomes even more significant.

## H.3 Limitations

We discuss the limitation of the proposed KP-GNN from two aspects.

**Stability**: Using K-hop instead of 1-hop can make the receptive field of a node increase with respect to $K$. For example, to compute the representation of a node with $L$ layer $K$-hop, GNN, the node will get information from all $LK$-hop neighbors. The increased receptive field can hurt learning, as mentioned in GINE+ [16]. It is an intrinsic limitation that exists in all $K$-hop GNNs. GINE+ [16] proposed a new way to fix the receptive field as $L$ by only considering $L - i$ layer representation of neighbor in $i$ hop during the aggregation. We also apply this approach and it helps mitigate the issue and shows great practical performance gain. Further, we propose a variant of KP-GNN named KP-GNN′, which only run KP-GNN at the first layer but 1-hop message passing at the rest of the layers. KP-GNN′ help mitigates stability issue and can be applied to large $K$ without causing the training of the model to fail. It achieves great results in various real-world datasets.

**Time complexity**: As we show above, we need $O(n^2)$ time complexity for KP-GNN, which is much higher than $O(m)$ of MPNN. However, this limitation exists for all subgraph-based methods like NGNN [26], GNN-AK [27], and ESAN [28] as they all require $O(nm)$ time complexity [23]. This is the sacrifice for better expressive power.

# I Implementation detail of KP-GNN

In this section, we discuss the implementation detail of KP-GNN.

**Combine function**: 1-hop message passing GNNs do not have COMBINE$^l$ function. Here we introduce two different COMBINE$^l$ functions. The first one is the attention [57] based combination mechanism, which automatically learns the importance of representation for each node at each hop. The second one uses the well-known geometric distribution [14]. The weight of hop $i$ is computed based on $\theta_i = \alpha(1 - \alpha)^i$, where $\alpha \in (0, 1]$. The final representation is calculated by weighted summation of the representation of all the hops. In our implementation, the $\alpha$ is learnable and different for each feature channel and each hop.

**Peripheral subgraph information**: In the current implementation, KP-GNN will compute two pieces of information. The first one is the peripheral edge set $E(Q_{v,G}^{k,t})$. In our implementation, we compute the number of edges for each distinct edge type. The second one is $k'$ configuration $C_k^{'k'}$, which contains node configuration and peripheral edges for all nodes in the peripheral subgraph. this is equivalent to running 1 layer of KP-GNN on each peripheral subgraph. Specifically, for node configuration, we compute the node configuration for each node and sum them up. For the peripheral edge set, we compute the total number of edges across all hops (do not consider edge type here). Finally, these two pieces of information are combined as $C_k^{'k'}$. All these steps are down in the preprocessing stage.

**KP-GCN, KP-GIN, and KP-GraphSAGE**: We implement KP-GCN, KP-GIN, and KP-GraphSAGE using the message and update function defined in GCN [1], GIN [7], and GraphSAGE [3] respectively. In each hop, independent parameter sets are used and the computation strictly follows the corresponding model. However, increasing the number of $K$ will also increase the total number of parameters, which is not scalable to $K$. To avoid this issue, we design the $K$-hop message passing in the following way. Suppose the total hidden size of the model is $H$, the hidden size of each hop is $H/K$. In this way, the model size is still on the same scale even with large $K$.

**KP-GIN+**: In a normal $K$-hop message passing framework, all $K$-hop neighbors will be aggregated for each node. It means that, after $L$ layers, the receptive field of GNN is $LK$. This may cause the unstable of the training as unrelated information may be aggregated. To alleviate this issue, we adapted the idea from GINE+ [16]. Specifically, we implement KP-GIN+, which applies exactly the same architecture as GINE+ except here we add peripheral subgraph information. At layer $l$, GINE+ only aggregates information from neighbors within $l$-hop, which makes a $L$ layer GINE+ still have a receptive field of $L$. Note that in KP-GIN+, we use a shared parameter set for each hop.

**KP-GIN$'$**: In KP-GIN$'$, we run a simple version of KP-GIN, which only uses KP-GIN at the first layer, but normal GIN for the rest of the layer. Although KP-GIN$'$ is weaker than KP-GIN from the expressiveness perspective. However, it is much more stable than KP-GIN in real-world datasets. Further, as we only have KP-GIN at the first layer, we can go larger $K$ without the cost of time complexity during training and inference. We observe great empirical results of KP-GIN$'$ on real-world datasets with less time complexity than normal KP-GNN.

**Path encoding**: To further utilize the graph structure information on each hop, we introduce the path encoding to KP-GNN. Specifically, we not only count whether two nodes are neighbors at hop $k$, but also count the number of walks with length $k$ between two nodes. Such information can be obtained with no additional cost as the $A^k$ of a graph $G$ with adjacency $A$ is a walk-counter with length $k$. Then the information is added to the $AGG_k^{l,normal}$ function as additional features.

**Other implementation**: For all GNNs, we apply the Jumping Knowledge method [58] to get the final node representation. The possible methods include sum, mean, concatenation, last, and attention. Batch normalization is used after each layer. For the pooling function, we implement mean, max, sum, and attention and different tasks may use different readout function.

## J  Experiential setting details

**EXP dataset**: For both K-GIN and KP-GIN, we use a hidden size of $48$. The final node representation is output from the last layer and the pooling method is the summation. In the experiment, we use 10-fold cross-validation. For each fold, we use 8 folds for training, 1 fold for validation, and 1 fold for testing. We select the model with the best validation accuracy and report the mean results across all folds. The training epoch is set to 40. In this experiment, we do not use path encoding for a fair

comparison. The learning rate is set to 0.001 and we use *ReduceLROnPlateau* learning rate scheduler with patience of 5 and a reduction factor of 0.5.

**SR25 dataset**: For both K-GIN and KP-GIN we use a hidden size of $64$. The final node representation is output from the last layer and the pooling method is the summation. For SR25 dataset, we directly train the validate the model on the whole dataset and report the best performance across 200 epochs. For each fold, we use 8 folds for training, 1 fold for validation, and 1 fold for testing. We select the model with the best validation accuracy and report the mean results across all folds. The training epoch is set to 200. In this experiment, we do not use path encoding for a fair comparison. The learning rate is set to 0.001.

**CSL dataset**: For both K-GIN and KP-GIN we use a hidden size of $48$. The final node representation is output from the last layer and the pooling method is the summation. In the experiment, we use 10-fold cross-validation.

**Graph&Node property dataset**: For graph and node property prediction tasks, we train models with independent 4 runs and report the mean results. For both K-GIN+ and KP-GIN+, the hidden size of models is set as 96. The final node representation is the concatenation of each layer. The pooling method is attention for graph property prediction tasks and sum for node property prediction tasks. The learning rate is 0.01 and we use *ReduceLROnPlateau* learning rate scheduler with patience of 10 and a reduction factor of 0.5. We use the shortest path distance kernel. The maximum number of epochs for each run is 250. For KP-GIN+, we search $K$ from 3 to 6 with/without path encoding and report the best result. For K-GIN+, we search $K$ from 3 to 6 without path encoding and report the best result.

**Graph substructure counting dataset**: For graph substructure counting tasks, we train models with independent 4 runs and report the mean results. For both K-GIN+ and KP-GIN+, the hidden size of models is set as 96. The final node representation is the concatenation of each layer. The pooling method is the summation. The learning rate is 0.01 and we use *ReduceLROnPlateau* learning rate scheduler with patience of 10 and a reduction factor of 0.5. We use the shortest path distance kernel. The maximum number of epochs for each run is 250. For KP-GIN+, we search $K$ from 1 to 4 with/without path encoding and report the best result. For K-GIN+, we search $K$ from 1 to 4 without path encoding and report the best result.

**TU datasets**: For TU datasets, we use 10-fold cross-validation. We report results for both settings in [7] and [36]. For the first setting, we use 9 folds for training and 1 fold for testing in each fold. After training, we average the test accuracy across all the folds. Then a single epoch with the best mean accuracy and the standard deviation is reported. For the second setting, we still use 9 folds for training and 1 fold for testing in each fold but we directly report the mean best test results. For KP-GNN, we implement GCN [1], GraphSAGE [3] and GIN [7] version. we search (1) the number of layer $\{2, 3, 4\}$, (2) the number of hop $\{2, 3, 4\}$, (3) the kernel of K-hop $\{spd, gd\}$, and (4) the $COMBINE$ function $\{attention, geometric\}$. The hidden size is 33 when $K = 3$ and 32 for the rest of the experiments. The final node representation is the last layer and the pooling method is the summation. The dropout rate is set as 0.5, the number of training epochs for each fold is 350 and the batch size is 32. The initial learning rate is set as $1e - 3$ and decays with a factor of 0.5 after every 50 epochs.

**QM9 dataset**: For QM9 dataset, we implement KP-GIN+ and KP-GIN$'$. For both two models, the hidden size of the model is 128. The final node representation is the concatenation of each layer and the pooling method is attention. The dropout rate is 0. For KP-GIN+, we use the shortest path distance kernel with $K = 8$ and 8 layers. Meanwhile, we add the virtual node to the model. For KP-GIN$'$, we use the shortest path distance kernel with $K = 16$ and 16 layers. Meanwhile, we add the residual connection. The maximum number of the peripheral edge is 6 and the maximum number of the component is 3. We also use additional path encoding in each layer. The learning rate is 0.001 and we use *ReduceLROnPlateau* learning rate scheduler with patience of 5 and a reduction factor of 0.7. If the learning rate is less than $1e - 6$, the training is stopped.

**ZINC datraset**: For ZINC dataset, we run the experiment 4 times independently and report the mean results. For each run, the maximum number of epochs is 500 and the batch size is 64. We implement the KP-GIN+ and KP-GIN$'$ for the ZINC dataset. For KP-GIN+, the hidden size is 104. The number of hops and the number of layers are both 8. For KP-GIN$'$, the hidden size is 96. The number of hops and the number of layers are 16 and 17 respectively. For both two models, we add the residual

Table 7: Ablation study on counting substructure dataset. (* means add path encoding.)

| model | K | Counting substructures (MAE) | | | |
|---|---|---|---|---|---|
| | | Triangle | Tailed Tri. | Star | 4-Cycle |
| **K-GIN+** | 1 | $0.4546 \pm 0.0107$ | $0.3665 \pm 0.0004$ | $0.0412 \pm 0.0468$ | $0.3317 \pm 0.0121$ |
| | 2 | $0.2938 \pm 0.0030$ | $0.2283 \pm 0.0012$ | $0.0206 \pm 0.0095$ | $0.2330 \pm 0.0020$ |
| | 3 | $0.2663 \pm 0.0088$ | $0.1998 \pm 0.0022$ | $0.0174 \pm 0.0030$ | $0.2202 \pm 0.0061$ |
| | 4 | $0.2593 \pm 0.0055$ | $0.1930 \pm 0.0033$ | $0.0165 \pm 0.0041$ | $0.2079 \pm 0.0024$ |
| **K-GIN+\*** | 1 | $0.4546 \pm 0.0107$ | $0.3665 \pm 0.0004$ | $0.0412 \pm 0.0468$ | $0.3317 \pm 0.0121$ |
| | 2 | $0.0132 \pm 0.0025$ | $0.0189 \pm 0.0049$ | $0.0219 \pm 0.0045$ | $0.0401 \pm 0.0027$ |
| | 3 | $0.0134 \pm 0.0020$ | $0.0147 \pm 0.0017$ | $0.0288 \pm 0.0062$ | $0.0471 \pm 0.0033$ |
| | 4 | $0.0253 \pm 0.0085$ | $0.0244 \pm 0.0028$ | $0.0171 \pm 0.0035$ | $0.0474 \pm 0.0025$ |
| **KP-GIN+** | 1 | $0.0060 \pm 0.0008$ | $0.0073 \pm 0.0020$ | $0.0151 \pm 0.0022$ | $0.2964 \pm 0.0080$ |
| | 2 | $0.0106 \pm 0.0015$ | $0.0115 \pm 0.0017$ | $0.0264 \pm 0.0064$ | $0.0657 \pm 0.0034$ |
| | 3 | $0.0134 \pm 0.0020$ | $0.0147 \pm 0.0017$ | $0.0288 \pm 0.0062$ | $0.0471 \pm 0.0033$ |
| | 4 | $0.0125 \pm 0.0012$ | $0.0169 \pm 0.0028$ | $0.0362 \pm 0.0113$ | $0.0761 \pm 0.0135$ |
| **KP-GIN+\*** | 1 | $0.0068 \pm 0.0019$ | $0.0083 \pm 0.0019$ | $0.0166 \pm 0.0041$ | $0.3063 \pm 0.0251$ |
| | 2 | $0.0110 \pm 0.0016$ | $0.0110 \pm 0.0016$ | $0.0255 \pm 0.0056$ | $0.0395 \pm 0.0018$ |
| | 3 | $0.0117 \pm 0.0024$ | $0.0111 \pm 0.0025$ | $0.0338 \pm 0.0077$ | $0.0813 \pm 0.0136$ |
| | 4 | $0.0155 \pm 0.0037$ | $0.0168 \pm 0.0018$ | $0.0422 \pm 0.0121$ | $0.0538 \pm 0.0052$ |

connection. The final node representation is the concatenation of each layer and the pooling method is the summation. We use the shortest path distance kernel. The initial learning rate is 0.001 and we use *ReduceLROnPlateau* learning rate scheduler with patience of 10 and a reduction factor of 0.5. If the learning rate is less than $1e - 6$, the training is stopped.

# K   Additional results

In this section, we provide additional experimental results and discussion. The additional results on the counting substructure dataset are shown in Table 7. First, for K-GIN+, we observe a steady improvement for all tasks when we increase the K, which aligns with the theoretical results. Second, path encoding can hugely boost the performance of the counting substructure, which demonstrates the effectiveness of path encoding. Third, for KP-GIN+, most of the tasks achieve the best result with only K=1. This means we only need local peripheral subgraph information to count through substructures. Even though increasing the K would increase the expressive power monotonically from a theoretical point of view, it may add noise to the training process. Finally, path encoding does not show much effect on the KP-GNN model, which means information on path encoding is already encoded in the peripheral subgraph.

# L   Datasets Description and Statistics

Table 8: Dataset statistics.

| Dataset | #Tasks | # Graphs | Ave. # Nodes | Ave. # Edges |
|---|---|---|---|---|
| EXP | 2 | 1200 | 44.4 | 110.2 |
| SR25 | 15 | 15 | 25 | 300 |
| CSL | 10 | 150 | 41.0 | 164.0 |
| Graph&Node property | 3 | 5120/640/1280 | 19.5 | 101.1 |
| Substructure counting | 4 | 1500/1000/2500 | 18.8 | 62.6 |
| MUTAG | 2 | 188 | 17.93 | 19.79 |
| D&D | 2 | 1178 | 284.32 | 715.66 |
| PTC-MR | 2 | 344 | 14.29 | 14.69 |
| PROTEINS | 2 | 1113 | 39.06 | 72.82 |
| IMDB-B | 2 | 1000 | 19.77 | 96.53 |
| QM9 | 12 | 129433 | 18.0 | 18.6 |
| ZINC | 1 | 10000 / 1000 / 1000 | 23.1 | 49.8 |