# OpenReview forum: "How Powerful are K-hop Message Passing Graph Neural Networks"
_NeurIPS.cc/2022/Conference — NeurIPS 2022 Accept_

### Official Review · Reviewer_uewD · 2022-07-10

**Rating:** 4
**Confidence:** 3
**Soundness:** 3 good
**Presentation:** 3 good
**Contribution:** 2 fair

**Summary:**

This paper provides a theoretical analysis of the expressive power of K-hop message passing GNNs. In particular, the authors show that K-hop message passing GNNs are more powerful than 1-hop message passing GNNs and that a K-hop GNN can distinguish some regular graphs while they are indistinguishable for 1-hop GNNs. Going further, they propose another message-passing GNN called KP-GNN, which is more powerful than K-hop GNNs. KP-GNNs can distinguish between many distance-regular graphs by using subgraph (edge)  information from the K-hops in addition to the node information. The authors also provide experiments to empirically verify the expressiveness of the newly proposed architectures for small K greater than 1. They also give some runtime analysis to show the computational cost of the proposed techniques.

**Questions:**

Please refer to the section on Weaknesses.

**Limitations:**

The authors have adequately addressed the limitations and potential negative societal impact of their work.

**Strengths And Weaknesses:**

Strengths
- The paper is theoretically sound, and the experiments give an idea about the effectiveness of the proposed techniques in graphs of practical interest.

Weakness
- While the paper shows that K-hop message passing GNNs are more powerful than 1-hop message passing GNNs and KP-GNNs are even more powerful than K-GNNs in expressiveness, the analysis doesn't provide an algorithm to check when a certain K suffices. It also doesn't tell us how these GNNs fit into established hierarchies. It is not clear how to use this analysis to inform practical applications.

- It is also not clear from the analysis if these K-hop message-passing networks are less/more powerful or incomparable to other proposed methods that use graph substructures like CW complexes, and subgraph structures.

- While the experiments show that the average values for KP-GNN-like models are higher the high variance in some tasks makes it unclear if they are much better than previous methods.

---

> ### Author Response · Authors · 2022-08-02
> **Author Response**
>
> ### **Response to reviewer uewD**
> Thanks for your insightful comments and reviews. We reply to all your comments below.
>
> > **Weakness 1**:
> While the paper shows that K-hop message passing GNNs are more powerful than 1-hop message passing GNNs and KP-GNNs are even more powerful than K-GNNs in expressiveness, the analysis doesn't provide an algorithm to check when a certain K suffices. It also doesn't tell us how these GNNs fit into established hierarchies. It is not clear how to use this analysis to inform practical applications.
>
>
> **A1**: The $K$ is a hyper-parameter that can be chosen flexibly given different tasks. To experimentally analyze the effect of $K$ on expressive power, we conduct the ablation study using EXP dataset shown in Table 2. From the results we can see that increasing the $K$ could increase the expressive power, which is also intuitive as increasing $K$ will include more distance and peripheral subgraph information. However, it does not always brings performance gains on real-world datasets. Thus, it is better to regard $K$ as a hyper-parameter.
> To further analyze the expressive power of $K$-hop message passing, we utilize recent results on 3-IGN to prove that **$K$-hop message passing is bounded by 3-WL.** But KP-GNN can surpass 3-WL on distance regular graphs. The details are in **A2** to reviewer qmM3.
>
>
> > **Weakness 2**:
> It is also not clear from the analysis if these K-hop message-passing networks are less/more powerful or incomparable to other proposed methods that use graph substructures like CW complexes, and subgraph structures.
>
> **A2**: (1) In **point 1** of the summary, we compare the $K$-hop message passing with Distance Encoding [1] and show that they share some commons but are actually incomparable from each other. (2) Inspired by recent results from [2], we further proved that $K$-hop message passing is bounded by 3-WL, which is the same as subgraph-based GNNs. (3) In **point 2** of the summary, we further compare KP-GNN with subgraph-based GNNs and show that **KP-GNN is strictly more powerful than subgraph-based GNNs in terms of distinguishing distance regular graph**. Thus, KP-GNN is not bounded by 3-WL. (4) In **point 3** of the summary, we show that the KP-GNN is both more space- and time-efficient than subgraph-based GNNs.
> SIN [3] and CW network [4] utilize the complex concept to perform high-order message passing on not the node but the complex. These methods are more related to GSN [5], and AutoBahn [6]. We leave the comparison for future work.
>
> > **Weakness 3**:
> While the experiments show that the average values for KP-GNN-like models are higher the high variance in some tasks makes it unclear if they are much better than previous methods.
>
> **A3**: The KP-GNN indeed has relatively higher variance in TU datasets compared to other methods. However, it is mainly due to the increase of the receptive field, which is an intrinsic limitation of all $K$-hop methods and we give a detailed discussion in response **A8** to reviewer 1P51. Meanwhile, the small size of the TU datasets also deteriorates this issue. However, by introducing the technique provided in [7], we actually achieve even more stable performance in ZINC and MolHiv datasets than other methods, as shown in Tables 5 and 6.
>
> Reference:
>
> [1] Pan Li, Yanbang Wang, Hongwei Wang, and Jure Leskovec. Distance encoding–design provably more powerful gnns for structural representation learning. Advances in Neural Information Processing Systems 33 (2020): 4465-4478.
>
> [2] Fabrizio Frasca, Beatrice Bevilacqua, Michael Bronstein, and Haggai Maron. Understanding and extending subgraph gnns by rethinking their symmetries. ArXiv, abs/2206.11140, 2022.
>
> [3] Cristian Bodnar, Fabrizio Frasca, Yu Guang Wang, Nina Otter, Guido Montufar, Pietro Liò,and Michael M. Bronstein. Weisfeiler and lehman go topological: Message passing simplicial networks. In ICLR 2021 Workshop on Geometrical and Topological Representation Learning,2021.
>
> [4] Cristian Bodnar, Fabrizio Frasca, Nina Otter, Yu Guang Wang, Pietro Liò, Guido Montufar, and Michael M. Bronstein. Weisfeiler and lehman go cellular: CW networks. In A. Beygelzimer,Y. Dauphin, P. Liang, and J. Wortman Vaughan, editors, Advances in Neural Informat Processing Systems, 2021.
>
> [5] Giorgos Bouritsas, Fabrizio Frasca, Stefanos Zafeiriou, and Michael M. Bronstein. Improving graph neural network expressivity via subgraph isomorphism counting, 2021.
>
> [6] Erik Henning Thiede, Wenda Zhou, and Risi Kondor. Autobahn: Automorphism-based graph neural nets. In A. Beygelzimer, Y. Dauphin, P. Liang, and J. Wortman Vaughan, editors,Advances in Neural Information Processing Systems, 2021.
>
> [7] Rémy Brossard, Oriel Frigo, and David Dehaene. Graph convolutions that can finally model local structure. arXiv preprint arXiv:2011.15069, 2020.

---

> > ### Author Response · Authors · 2022-08-08
> > **We look forward to your reply**
> >
> > We thank reviewer uewD again for the inspiring comments to help us improve the paper. In response to the comments, we address the main concerns as follows:
> > 1. **The effect of $K$**: we state that increasing the $K$ can improve the expressive power of the model, which is empirically verified by an ablation study on EXP dataset (Table 2). However, increasing $K$ does not always boot performance on real-world datasets. It is better to view $K$ as a hyper-parameter.
> > 2. **Comparison between $K$-hop message passing and other methods**: We compare the difference between $K$-hop message passing in **point 1** of the summary and also prove that it is bounded by 3-WL. We compare the proposed KP-GNN with subgraph-based GNNs in **point 2** of the summary and show that KP-GNN is more powerful than subgraph-based GNNs in terms of distinguishing distance regular graphs. In **point 3** of the summary, we point out KP-GNN has better time and space complexity than subgraph-based GNNs.
> > 3. **High variance**: We state that the high variance is a common limitation for all $K$-hop message passing and the size of TU dataset further deteriorates this issue. However, by introducing the technique proposed in [1], we achieve even lower variance than other methods on ZINC and OGB datasets.
> >
> >
> > We valued the reviewer's feedback and made a great effort in writing the author's response. Since there are about 1 days left in the discussion phase, would you mind letting us know if our response addresses your concern? If you think there are still other issues, please kindly let us know, we are happy to follow up with you before the discussion phase ends.
> >
> > [1] Rémy Brossard, Oriel Frigo, and David Dehaene. Graph convolutions that can finally model local structure. arXiv preprint arXiv:2011.15069, 2020.

---

### Official Review · Reviewer_QvfD · 2022-07-10

**Rating:** 3
**Confidence:** 4
**Soundness:** 2 fair
**Presentation:** 3 good
**Contribution:** 2 fair

**Summary:**

This work first formalizes the expressive power of the K-hop message passing GNNs. Then, it proposes KP-GNN, which uses the peripheral subgraph information, to improve the expressiveness of K-hop message passing GNNs. Experiments on various graph tasks are performed to evaluate the proposed KP-GNN.

**Questions:**

As in the above weaknesses.

**Limitations:**

The main limitation is that the time complexity of KP-GNN is quite high, which might be not scalable to large graphs.

**Strengths And Weaknesses:**

++++ Strengths

(1) The unified view of K-hop message passing GNNs (Figure 1) is clear and potentially useful to the community.

(2) The presentation of this paper is well organized and easy to follow.

++++ Weaknesses

The main weakness of this paper is that the contribution is not clearly stated, the key differences with prior works are not clear, some of the claims are not clarified well. The details are as follows.

(1) The expressive power of k-hop message passing GNNs are not clearly presented. Please correct me if I am understanding it wrongly. In Definition 3, we have a proper k-hop message passing GNNs if all three functions are injective. Then, in Definition 4, the node configuration is a list, which means it has an order. I think the gap from Definition 3 to Definition 4 has not been clearly stated. To my understanding, an injective COMBINE function in Eq. (3) does not guarantee the order of hops. For example, for two node v_1, v_2, let’s say the input to the COMBINE function is {{1}} and {{2}} if we only consider 1 hop. Then, when we consider 3 hops, an example input of the COMBINE function could be {{1,2,3}} and {{2,3,1}}. Then, the COMBINE function will yield the same output for these two nodes.

In other orders, in Collorary 1, the statement “the K-hop message passing is at least equally powerful as 1-hop message passing since the K-hop message passing includes all the information that 1-hop message passing has” may not hold, since the 1-hop information may be destroyed when we COMBINE it with more information from more hops.

(2) The difference to GNN-AK and Nested GNN has not been clearly made. In these two works, they consider encoding the k-hop subgraphs for all nodes, which is quite related to the peripheral subgraph information considered in KP-GNN. The difference and advantages over these works are not clear in the current paper. For example, technically, how powerful of KP-GNN compared to these works? In the current manuscript, we cannot see significant improvements over these works in terms of empirical performance or time complexity.

---

> ### Author Response · Authors · 2022-08-02
> **Author Response 1/2**
>
> ### **Response to reviewer QvfD**
> Thank you for your review and suggestions. We reply to all your comments below.
>
> > **Weakness 1**:
> The expressive power of k-hop message passing GNNs are not clearly presented. Please correct me if I am understanding it wrongly. In Definition 3, we have a proper k-hop message passing GNNs if all three functions are injective. Then, in Definition 4, the node configuration is a list, which means it has an order. I think the gap from Definition 3 to Definition 4 has not been clearly stated. To my understanding, an injective COMBINE function in Eq. (3) does not guarantee the order of hops. For example, for two node v_1, v_2, let’s say the input to the COMBINE function is {{1}} and {{2}} if we only consider 1 hop. Then, when we consider 3 hops, an example input of the COMBINE function could be {{1,2,3}} and {{2,3,1}}. Then, the COMBINE function will yield the same output for these two nodes.In other orders, in Collorary 1, the statement “the K-hop message passing is at least equally powerful as 1-hop message passing since the K-hop message passing includes all the information that 1-hop message passing has” may not hold, since the 1-hop information may be destroyed when we COMBINE it with more information from more hops.
>
> **A1**: Indeed the injective COMBINE function does not guarantee the order of hops. However, the key point is that in $K$-hop message passing, we have different MES and UPD functions for each hop, as shown in Eq. (3). This can be regarded as the **positional encoding for different hops**. In other words, the input $h_v^{l,k}$ to the COMBINE function already contains the hop it belongs to. For your example, the actual input when considering 3 hops is not {{1,2,3}} and {{2,3,1}}, but rather {{$1^{1},2^{2},3^{3}$}} and {{$2^{1},3^{2},1^{3}$}}, where the superscript denotes the positional (hop) encoding injected by the different MES and UPD at different hops. Therefore, even using a multiset function COMBINE that is invariant to order, we can still distinguish the $h_v^{l,k}$ from different hops and will not mix their information, and **all our theoretical results remain established**.
>
>
> > **Weakness 2**:
> The difference between GNN-AK and Nested GNN has not been clearly made. In these two works, they consider encoding the k-hop subgraphs for all nodes, which is quite related to the peripheral subgraph information considered in KP-GNN. The difference and advantages over these works are not clear in the current paper. For example, technically, how powerful of KP-GNN compared to these works? In the current manuscript, we cannot see significant improvements over these works in terms of empirical performance or time complexity.
>
> **A2**:
> First, consider neighbor information from different hops is closely related to distance encoding [1]. However, in **point 1** of the summary, we show even distance encoding is not identical to $K$-hop message passing and neither of them is strictly more expressive than the other in node expressive power.
> Second, as pointed out in **point 2** of the summary, in $K$-hop message passing and KP-GNN, we only need to perform message passing on the whole graph and each node has only one representation. Instead, subgraph-based GNNs perform message passing on each subgraph and each node has multiple representations. KP-GNN is also different from subgraph-based GNNs like NGNN [2], GNNAK [3] in how they view the subgraph. Subgraph-based GNNs view the subgraph as a whole and directly apply vanilla message passing on it. Instead, by using $K$-hop message passing and peripheral subgraph information, the KP-GNN further divides the subgraph into two parts: the hierarchy of neighbors from different hops and the connection structure of neighbors at each hops. Further, we show that **KP-GNN is strictly more powerful than subgraph-based GNNs in terms of distinguishing distance regular graph**. This further distinguishes the KP-GNN from subgraph-based GNNs. Finally, in **point 3** of the summary, we re-analyze the time and space complexity and show that **KP-GNN is also more time- and space-efficient compared to subgraph-based GNNs**. We have included all the discussion and proof in the updated version.

---

> > ### Author Response · Authors · 2022-08-02
> > **Author Response 2/2**
> >
> >
> > > **limitation**:
> > The main limitation is that the time complexity of KP-GNN is quite high, which might be not scalable to large graphs.
> >
> > **A3**: As we show in **point 3** of the summary, the $K$-hop message passing and KP-GNN require higher time complexity than vanilla message passing. However, such limitation exists in all subgraph-based GNNs and is even worse. This is the sacrifice for better expressive power. Further, KP-GNN has much less time and space complexity than subgraph-based GNNs but can achieve higher expressiveness in distinguishing distance regular graphs.
> >
> >
> > Reference
> >
> > [1] Pan Li, Yanbang Wang, Hongwei Wang, and Jure Leskovec. Distance encoding–design provably more powerful gnns for structural representation learning. Advances in Neural Information Processing Systems 33 (2020): 4465-4478.
> >
> > [2] Muhan Zhang and Pan Li. Nested graph neural networks. In A. Beygelzimer, Y. Dauphin,P. Liang, and J. Wortman Vaughan, editors, Advances in Neural Information Processing Systems,2021.
> >
> > [3] Lingxiao Zhao, Wei Jin, Leman Akoglu, and Neil Shah. From stars to subgraphs: Uplifting any GNN with local structure awareness. In International Conference on Learning Representations,2022.

---

> > > ### Comment · Reviewer_QvfD · 2022-08-05
> > > **Concern with the COMBINE function**
> > >
> > > Thank you for clarifying the difference with prior works.
> > >
> > > I still do not think the **weakness 1** has been resolved. Yes, you can use different MES and UPD functions for each hop. However, the COMBINE function is not aware of the order of the hop. In other words, by looking at the output of the COMBINE function, we are not able to infer what the inputs are for the COMBINE function. Hence, it is not injective.

---

> > > > ### Author Response · Authors · 2022-08-05
> > > > **About the COMBINE function**
> > > >
> > > > Thank you for your response!
> > > >
> > > > Firstly, the COMBINE function can be an injective function of its input multiset, which is guaranteed by Deep Sets [1]. We believe what the reviewer concerns is whether from the output of COMBINE we can recover the aggregated messages $h_v^{l,k}$ at **different** hops $k$. This is actually feasible. Firstly, since COMBINE is an injective multiset function, we can recover the multiset  {{$h_v^{l,k} | k=1,2,...,K$}} from the output. Then, the only thing we need is to tell apart each $h$ term is from which hop. This is guaranteed by the distinct MES and UPD functions for each hop. You can understand them as first doing normal message passing to calculate an $\hat{h}_v^{l,k}$ for each hop, and then concatenating a unique hop ID $k$ to each $\hat{h}_v^{l,k}$ to form the final $h_v^{l,k}:= \text{Concat}(\hat{h}_v^{l,k}, k)$. The final dimension is only used to mark its hop. Therefore, from {{$h_v^{l,k} | k=1,2,...,K$}}, we can exactly recover the $h$ term for each hop by checking their final dimension. And since MES and UPD are implemented by MLPs, the above constructed function is learnable based on the universal approximation theorem.
> > > >
> > > > [1] Zaheer, Manzil, et al. "Deep sets." Advances in neural information processing systems 30 (2017).

---

> > > > > ### Comment · Reviewer_QvfD · 2022-08-05
> > > > > **Can you provide a formal proof**
> > > > >
> > > > > Yes, my concern is that "whether from the output of COMBINE we can recover the aggregated messages for different hops". According to the provided explanation, if we **fix** the final dimension of the MLP output, how can we ensure that MLP is a **universal** approximation theorem?
> > > > >
> > > > > I do think the current proof is informal. Could you provide a more formal and conniving proof? Thank you.

---

> > > > > > ### Author Response · Authors · 2022-08-05
> > > > > > **About MLP's universal approximation**
> > > > > >
> > > > > > Thanks for the additional question! We are glad to address the further concern. The [universal approximation theorem](https://en.wikipedia.org/wiki/Universal_approximation_theorem) states that given enough hidden dimensions, a two-layer MLP with certain weights can approximate any continuous function to an arbitrary small error. We are not fixing the final dimension of the MLP output by overriding its original output. Instead, we construct such a function (which outputs the hop number $k$ at its final output dimension), and rely on MLP's universal approximation ability to approximate such a function to an arbitrary small error. This is standard in many previous works proving a network's expressive power, such as [1] and [2].
> > > > > >
> > > > > > And such an MLP indeed exists. Let the UPD$^l_k$ in (3) be an MLP. We let its second-last layer's weights connecting to the last layer's final output dimension be all zero (so that any output from the second-last layer results in zero after the linear transformation). Then we let the bias associated with the last layer's final output dimension be $k$. This way, we have constructed an MLP UPD$^l_k$ that always outputs $k$ at its final output dimension. Its previous dimensions are used for the normal purpose.
> > > > > >
> > > > > > The above **formally** proves the existence of such an MLP without relying on the universal approximation theorem. We hope that addresses your concern. In practice, the learned UPD$^l_k$ might not have exactly the same form, but it has various ways to implicitly encode the information of $k$ into its output so that different $h$ terms won't be mixed.
> > > > > >
> > > > > >
> > > > > > [1] Zaheer, Manzil, et al. "Deep sets." Advances in neural information processing systems 30 (2017).
> > > > > >
> > > > > > [2] Xu, Keyulu, et al. "How Powerful are Graph Neural Networks?." International Conference on Learning Representations. 2018.

---

> > > > > > > ### Comment · Reviewer_QvfD · 2022-08-06
> > > > > > > **Further concern about the proof**
> > > > > > >
> > > > > > > Thank you for the response.
> > > > > > >
> > > > > > > Yes, you can construct such a function (which outputs the hop number $k$ at its final output dimension), and rely on MLP's universal approximation ability to approximate such a function to an arbitrary small error. However, this means that you already spent one dimension of the MLP output to encode the hop information (explicitly or implicitly). Then the output of you MLP UPD cannot achieve universal function over the input, since one dimension has to be used for encoding hop information. Please correct me if I am understanding wrongly.
> > > > > > >
> > > > > > > By the way, how is the UPD function implemented/parameterized in details.

---

> > > > > > > > ### Author Response · Authors · 2022-08-06
> > > > > > > > **Clarifications**
> > > > > > > >
> > > > > > > > Thanks for the question. `"The output of you MLP UPD cannot achieve universal function over the input, since one dimension has to be used for encoding hop information"` is actually not correct. The universal approximation property still holds for our MLP, since we are **not** using a special MLP whose last dimension is fixed to $k$, but use a **standard MLP with free parameters** to learn to approximate the function. Suppose the function we want our MLP to learn is the one that encodes the normal $k$-hop neighbor information into its first $d-1$ output dimensions and encodes the hop information $k$ into its $d$-th output dimension. Since such a function (by construction) exists, then the universal approximation theorem guarantees that an MLP approximating the function arbitrarily well also exists (with its parameters set to certain values). Note that, the universal approximation theorem **only** guarantees the **existence** of such MLPs, but does not tell us how to find the proper parameters. Modern deep learning leverages stochastic versions of gradient descent and almost always demonstrates near-perfect approximation ability of continous functions.
> > > > > > > >
> > > > > > > > As for whether the MLP's universal approximation of the first $d-1$ output dimensions will be affected by its encoding of hop information (in the last dimension), the answer is still no, since different output dimensions have separate weights connecting from the second-last layer. Suppose now we don't need to encode the hop information, then there must exist an MLP1 that approximates the first $d-1$ dimensions to an arbitrary small error. Then, when we want to additionally encode the hop information, we can modify MLP1 by 1) adding another output dimension, 2) making all the weights connecting to the added dimension zero, 3) making the bias connecting to the added dimension $k$, and 4) keeping all the remaining parts unchanged. Such a new MLP2 does not affect MLP1's universal approximation of the first $d-1$ dimensions. Further, it encodes the hop $k$ in its $d$-th output dimension.
> > > > > > > >
> > > > > > > > In our implementation, we use the same UPD function as in GIN [1], which is exactly an MLP over the summed message and previous-layer node representation. It is a standard MLP without fixing any dimension.
> > > > > > > >
> > > > > > > > [1] Xu, Keyulu, et al. "How Powerful are Graph Neural Networks?." International Conference on Learning Representations. 2018.

---

> > > > > > > > > ### Comment · Reviewer_QvfD · 2022-08-08
> > > > > > > > > **A formal proof is needed**
> > > > > > > > >
> > > > > > > > > Thank you for your response. I am still having the concern about the "hop information" encoding of the COMBINE function. The current explanation is informal and not valid to me. I recommend writing a formal proof in the next version.

---

> > > > > > > > > > ### Author Response · Authors · 2022-08-08
> > > > > > > > > > **A formal proof**
> > > > > > > > > >
> > > > > > > > > > We thank the reviewer for the suggestion of adding a formal proof. We include the proof in Appendix K of the updated version.
> > > > > > > > > >
> > > > > > > > > > To prove that the ouput of COMBINE function can recover the hop information is equivalent to prove that the **output of Equation (3) is an injective mapping of the neighbor representations at different hops**. However, there are issues on directly formatting many equations in Openreview thus we don't include the proof in the response. We are sorry for the inconvenience. Please see the Appendix K for the detail proof.
> > > > > > > > > >
> > > > > > > > > > Hope this formal proof could address your concern.

---

### Official Review · Reviewer_1P51 · 2022-07-11

**Rating:** 6
**Confidence:** 4
**Soundness:** 3 good
**Presentation:** 3 good
**Contribution:** 3 good

**Summary:**

The authors theoretically analyze the expressive power of existing K-hop message passing graph neural networks (K-GNNs). Furthermore, the authors propose the K-hop peripheral-subgraph-enhanced graph neural network (KP-GNN), which aggregates peripheral subgraph information to enhance the expressive power of K-GNNs. Experiments demonstrate that KP-GNN achieves competitive results on all benchmark datasets.

**Questions:**

1. How are the shortest path distance kernel and graph diffusion kernel misused in existing works?
2. Why is the increase of the computational costs in terms of $K$ far less than the theoretical results?
3. Besides the running time at each epoch, the authors may want to directly use the total running time to evaluate the efficiency, as the total running time depends on not only the running time at each epoch but also the convergence rate.
4. The authors may want to use the multiset to define the K-hop neighbors rather than the set in Definition 2. Following Definition 2 in the main text, $\mathcal{N}_{v,G}^{2,gd}$ of node 2 in the example in Figure 2 in the appendix is $\{v_1,3,4,5,6\}$ rather than the multiset $\{v_1,3,3,4,4,5,6\}$.
5. Some important experiment settings are missing, including times of each experiment and the definition of error bars in Tables 3, 5, 6.


**Limitations:**

The authors may want to address the limitations of the proposed method, e.g. types of data or tasks the method might fail.

**Strengths And Weaknesses:**

Strengths:
1. The authors provide the theoretical analysis for the expressive power of K-GNNs and KP-GNN.
2. Experiments demonstrate that KP-GNN achieves competitive results on all benchmark datasets.
3. The authors provide many typical examples so that the derivation of KP-GNN is easy to follow.

Weaknesses:

My major concerns are follows.
1. The authors assume that all nodes in the graph have the same feature in the theoretical analysis, which is stronger than the assumption in [1]. The authors may want to analyze the expressive power of KP-GNN under the assumption of [1].
2. The authors claim that the expressive power of KP-GNN outperforms existing K-GNNs. Therefore, the authors may want to compare KP-GNN with more K-GNNs in experiments, such as MixHop [2], K-hop [3], MA-GNA [4], GPR-GNN [5].
3. The authors may want to compare the time and space complexity of KP-GNN with other powerful GNNs such as PPGN [6].

- [1] Xu Keyulu, Weihua Hu, Jure Leskovec and Stefanie Jegelka. "How Powerful are Graph Neural Networks?" International Conference on Learning Representations. 2018.
- [2] Sami Abu-El-Haija, Bryan Perozzi, Amol Kapoor, Nazanin Alipourfard, Kristina Lerman, Hrayr Harutyunyan, Greg Ver Steeg and Aram Galstyan. "Mixhop: Higher-Order Graph Convolutional Architectures via Sparsified Neighborhood Mixing." International Conference on Machine Learning. 2019.
- [3] Nikolentzos, Giannis, George Dasoulas, and Michalis Vazirgiannis. "K-Hop Graph Neural Networks." Neural Networks. 2020.
- [4] Guangtao Wang, Rex Ying, Jing Huang and Jure Leskovec. "Multi-hop Attention Graph Neural Networks." IJCAI. 2021.
- [5] Eli Chien, Jianhao Peng, Pan Li and Olgica Milenkovic. "Adaptive Universal Generalized PageRank Graph Neural Network." International Conference on Learning Representations. 2020.
- [6] Haggai Maron, Heli Ben-Hamu, Hadar Serviansky and Yaron Lipman. "Provably Powerful Graph Networks." Advances in Neural Information Processing Systems. 2019.

---

> ### Author Response · Authors · 2022-08-02
> **Author Response 1/2**
>
> ### **Response to reviewer 1P51**
> We thank the reviewer for the insightful comments and suggestions. We reply to all your comments below.
>
> > **Weakness 1**:
> > The authors assume that all nodes in the graph have the same feature in the theoretical analysis, which is stronger than the assumption in [1]. The authors may want to analyze the expressive power of KP-GNN under the assumption of [1].
>
> **A1**: In [1], the author assumes the features belong to a countable universe, which means that the input feature could be different for different nodes. This assumption is weaker than our assumption. Actually, our assumption analyzes the lower bound of the expressive power of a model as adding distinct node features will only increase the expressive power. For example, consider example 2 in figure 1 of our paper, if we assume all node features are 1 except node 4 is 2 in both two graphs. Then, considering the 2-height rooted subtree of node $v_1$ and $v_2$, we can see they are different as the rooted subtree of $v_1$ contains nodes labeled with 2 but it is not true for $v_2$. This means that we can use 1-WL algorithm with only 2 iterations or MPNN with only two layers to distinguish the two nodes, which is previously undistinguishable.
> Furthermore, our assumption is natural and follows previous works like NGNN [7] and Distance Encoding [8], which explicitly makes the same assumption. Meanwhile, GNN-AK [9], GraphSNN [10] also implicitly uses this assumption when illustrating their methods and theories.
>
> > **Weakness 2**:
> The authors claim that the expressive power of KP-GNN outperforms existing K-GNNs. Therefore, the authors may want to compare KP-GNN with more K-GNNs in experiments, such as MixHop [2], K-hop [3], MA-GNA [4], GPR-GNN [5].
>
> **A2**:
> 1.From the theoretical point of view, all models with $K$-hop message passing framework have the same expressive power. We have detailed discussion on existing K-hop methods in Appendix B. In fact, MixHop [2], MA-GNA [4], and GPR-GNN [5] all use a weak version of $K$-hop message passing.
> 2.The **K-GIN+** in table 1 of the main paper exactly follows the $K$-hop GNN framework using shortest path distance kernel and KP-GNN shows better performance in almost all tasks. Here we provide additional results to compare KP-GNN with GPR-GNN and K-GIN+ with graph diffusion kernel on substructure counting dataset.
>
> **Counting Substructure**
> | Model  | Tri. | Tailed Tri. | Star |4-Cycle|
> |:---:|:---:|:---:|:--:|:--:|
> |**GPR-GNN**|0.4690|0.4389|0.9723|0.3874|
> |**K-GIN+(gd)**|0.2258|0.1279|0.0463|0.0539|
> |**K-GIN+(spd)**|0.1180|0.0747|**0.0009**|0.0840|
> |**KP-GIN+**|**0.0377**|**0.0314**|0.0024|**0.0258**|
>
>
> We can see KP-GNN still outperforms all methods.
>
>
> > **Weakness 3**:
> The authors may want to compare the time and space complexity of KP-GNN with other powerful GNNs such as PPGN [6].
>
> **A3**: From the **point 3** of the summary, the space and time complexity of KP-GNN are $O(n)$ and $O(n^2)$ respectively, which is far less than PPGN[6], which has $O(n^2)$ and $O(n^3)$ space and time complexity.
>
> > **Question 1**:
> How are the shortest path distance kernel and graph diffusion kernel misused in existing works?
>
> **A4**: For example, MixHop [1], K-hop [2], MA-GNA [4], GPR-GNN [5], and GINE+ [11] all claim they utilize information from more than 1-hop and can be classified into $K$-hop based GNNs. However, based on our definition, GINE+ and K-hop use the shortest path distance kernel but MixHop, MA-GNA, GPR-GNN assume the graph diffusion kernel with diffusion value as pre-defined weight in aggregation. They have different expressive power based on our analysis and should be classified into different categories.
>
> > **Question 2**:
> Why is the increase of the computational costs in terms of  far less than the theoretical results?
>
> **A5**: As graphs are sparse in most cases and we only use small $K$, the real computation cost added is much smaller than the theoretical result.

---

> > ### Author Response · Authors · 2022-08-02
> > **Author Response 2/2**
> >
> >
> > > **Question 3**:
> > Besides the running time at each epoch, the authors may want to directly use the total running time to evaluate the efficiency, as the total running time depends on not only the running time at each epoch but also the convergence rate.
> >
> > **A6**: As you mentioned, the total running time depends on the convergence rate. However, clear identification of the time of convergence is hard and it is different even for different tasks. This makes the use of total running time for analysis and comparison impractical. Instead, normally the deep learning model is trained with fixed epochs even if it is already convergent. Therefore, we think analyzing the epoch time is more reasonable.
> >
> > > **Question 4**:
> > The authors may want to use the multiset to define the K-hop neighbors rather than the set in Definition 2. Following Definition 2 in the main text, $\mathcal{N}^{2,gd}_{v,G}$ of node 2 in the example in Figure 2 in the appendix is $v_1,3,4,5,6$ rather than the multiset $v_1,3,3,4,4,5,6$.
> >
> > **A7**:  $\mathcal{N}^{k,t}_{v,G}$ means the set of all neighbors from 1 to $k$-th hops. However, in Figure 2, it is not the  $\mathcal{N}^{2,gd}_{v,G}$ but rather $Q^{1,gd}_{v,G} \cup Q^{2,gd}_{v,G}$.
> >
> > > **Question 5**:
> > Some important experiment settings are missing, including times of each experiment and the definition of error bars in Tables 3, 5, 6.
> >
> > **A8**:For experiment times, we include the details in the appendix. The error bars mean the variance of multiple runs of experiments. We added the definition of error bars in the updated version.
> >
> > > **Limitation**:
> > The authors may want to address the limitations of the proposed method, e.g. types of data or tasks the method might fail.
> >
> > **A8**: We discuss the limitations of KP-GNN from two aspects. (1) **stability**: Using K-hop instead of 1-hop can make the receptive field of a node increase with respect to $K$. For example, to compute the representation of a node with $L$ layer $K$-hop GNN, the node will get information from all $LK$-hop neighbors. The increased receptive field can hurt learning, as mentioned in GINE+ [11]. It is an intrinsic limitation that exists in all $K$-hop based methods. GINE+ [11] proposed a new way to fix the receptive field to $L$ by only considering representation from the $L-i$-th layer of neighbor in $i$ hop during the aggregation. We also apply this approach and it helps mitigate the issue and shows great practical performance gain. (2)**Time complexity**: As we show in **point 3** of the summary, the $K$-hop message passing and KP-GNN require higher time complexity than vanilla message passing. However, such limitation exists and is even worse in all subgraph-based GNNs. This is the sacrifice for better expressive power. Further, KP-GNN has much less space complexity than subgraph-based GNNs but can achieve higher expressiveness on distinguihsing distance regular graphs.
> >
> >
> > Reference
> > [1] Xu Keyulu, Weihua Hu, Jure Leskovec and Stefanie Jegelka. "How Powerful are Graph Neural Networks?" International Conference on Learning Representations. 2018.
> >
> > [2] Sami Abu-El-Haija, Bryan Perozzi, Amol Kapoor, Nazanin Alipourfard, Kristina Lerman, Hrayr Harutyunyan, Greg Ver Steeg and Aram Galstyan. "Mixhop: Higher-Order Graph Convolutional Architectures via Sparsified Neighborhood Mixing." International Conference on Machine Learning. 2019.
> >
> > [3] Nikolentzos, Giannis, George Dasoulas, and Michalis Vazirgiannis. "K-Hop Graph Neural Networks." Neural Networks. 2020.
> > [4] Guangtao Wang, Rex Ying, Jing Huang and Jure Leskovec. "Multi-hop Attention Graph Neural Networks." IJCAI. 2021.
> > [5] Eli Chien, Jianhao Peng, Pan Li and Olgica Milenkovic. "Adaptive Universal Generalized PageRank Graph Neural Network." International Conference on Learning Representations. 2020.
> >
> > [6] Haggai Maron, Heli Ben-Hamu, Hadar Serviansky and Yaron Lipman. "Provably Powerful Graph Networks." Advances in Neural Information Processing Systems. 2019.
> >
> > [7] Muhan Zhang and Pan Li. Nested graph neural networks. In A. Beygelzimer, Y. Dauphin,P. Liang, and J. Wortman Vaughan, editors, Advances in Neural Information Processing Systems,2021.
> >
> > [8] Pan Li, Yanbang Wang, Hongwei Wang, and Jure Leskovec. Distance encoding–design provably more powerful gnns for structural representation learning. Advances in Neural Information Processing Systems 33 (2020): 4465-4478.
> >
> > [9] Lingxiao Zhao, Wei Jin, Leman Akoglu, and Neil Shah. From stars to subgraphs: Uplifting any GNN with local structure awareness. In International Conference on Learning Representations,2022.
> >
> > [10] Asiri Wijesinghe and Qing Wang. A new perspective on ”how graph neural networks go beyond weisfeiler-lehman?”. In International Conference on Learning Representations, 2022.
> >
> > [11] Rémy Brossard, Oriel Frigo, and David Dehaene. Graph convolutions that can finally model local structure. arXiv preprint arXiv:2011.15069, 2020.

---

> > > ### Comment · Reviewer_1P51 · 2022-08-07
> > > **Thanks for the authors' rebuttal**
> > >
> > > Thanks for the authors' rebuttal. I have read the authors' response and all the other reviewers' comments. However, my major concerns have not been properly addressed.
> > > 1. The authors may want to explain why the assumption in this paper is weaker than the assumption in [1] in the revised version.
> > > 2. The authors claim that the shortest path distance kernel and graph diffusion kernel are misused in existing works without an explanation (see Line 37) in the revised version.
> > > 3. The authors may want to explain the increase of the computational costs in terms of $K$ is far less than the theoretical results (see Line 350) in the revised version.
> > > 4. The total running time is useful to evaluate the efficiency. The authors may want to train models with early stopping rather than fixed epochs.
> > > 5. What are the $N_{v,G}^{2,gd}$ and $Q_{v,G}^{1,gd} \cup Q^{2,gd}_{v,G}$ of node 2 in the example in Figure 2?
> > > 6. The definition of the error bar is confusing. The authors claim that they use the variance as the error bar in the rebuttal, while Line 305 points out that they use the standard deviation as the error bar.
> > >
> > > [1] Xu Keyulu, Weihua Hu, Jure Leskovec and Stefanie Jegelka. "How Powerful are Graph Neural Networks?" International Conference on Learning Representations. 2018.

---

> > > > ### Author Response · Authors · 2022-08-07
> > > > **Further Response**
> > > >
> > > > Thanks for your response, we are glad to address your further concern:
> > > > >1.The authors may want to explain why the assumption in this paper is weaker than the assumption in [1] in the revised version.
> > > >
> > > > **A1**:We include the explanation in Line 105-106 of current version (Line 107-108 of updated version).
> > > >
> > > > >2.The authors claim that the shortest path distance kernel and graph diffusion kernel are misused in existing works without an explanation (see Line 37) in the revised version.
> > > >
> > > > **A2**: Thanks for your suggestion, we include explanation in the Line 37-39 of updated version.
> > > >
> > > > >3.The authors may want to explain the increase of the computational costs in terms of is far less than the theoretical results (see Line 350) in the revised version.
> > > >
> > > > **A3**: Thanks for your suggestion, we include explanation in the Line 352-353 of updated version.
> > > >
> > > > >4. The total running time is useful to evaluate the efficiency. The authors may want to train models with early stopping rather than fixed epochs.
> > > >
> > > > **A4** Thanks for your suggestion, we will include experiment of total running time in the future version.
> > > >
> > > > >5.What are the $N_{v,G}^{2,gd}$ and $Q_{v,G}^{1,gd} \cup Q_{v,G}^{2,gd}$ of node 2 in the example in Figure 2?
> > > >
> > > > **A5** $N_{2,G}^{2,gd}$ is $\\{ 3,4,5,6,v_1\\} $, $Q_{2,G}^{1,gd}$ is $\\{ 3,4,v_1\\}$, $Q_{2,G}^{2,gd}$ is $\\{3,4,5,6\\}$. We use $Q_{2,G}^{1,gd} \cup Q_{2,G}^{2,gd}$ in rebuttal because we want to illustrate why node 2 in figure 2 is that way. But we believe it may cause further confusion. Here we explain it from another view. In the aggregation of $K$-hop message passing, $Q_{2,G}^{1,gd}$ and $Q_{2,G}^{2,gd}$ will be aggregated using different set of parameters. It is the key reason why there are "repeated" nodes in the rooted subtree of node 2. However, we use different color for different hops to distinguish them. So there is no multiset in the Figure 2 but rather two independent set.
> > > >
> > > > >6.The definition of the error bar is confusing. The authors claim that they use the variance as the error bar in the rebuttal, while Line 305 points out that they use the standard deviation as the error bar.
> > > >
> > > > **A6**: We apologize for the confusion. The error bar is standard deviation as indicated in the Line 305 of current version(Line 306 of updated version).
> > > >
> > > > Again, thanks for your constructive comments and we include all the revision in the updated version except total running time experiments. We will conduct experiments and update results as soon as possible.

---

> > > > > ### Comment · Reviewer_1P51 · 2022-08-08
> > > > > **Thanks for the authors' rebuttal and their efforts to improve this work**
> > > > >
> > > > > Thanks for the authors' rebuttal and their efforts to improve this work. The response has addressed my questions. I will stick to my original score (6, weak accept). Best wishes.

---

### Official Review · Reviewer_qmM3 · 2022-07-18

**Rating:** 5
**Confidence:** 5
**Soundness:** 2 fair
**Presentation:** 3 good
**Contribution:** 2 fair

**Summary:**

The paper studies the K-hop message passing GNN. The paper shows that the original K-hop GNN is more powerful than 1-WL. To further improve its expressivity, the author further proposes to use the induced subgraph at each hop to augment the original K-hop GNN. interesting theoretical results are proposed for distinguishing regular graphs.

**Questions:**

Overall the paper studies the k-hop message passing in-depth and has done many experiments. I don't have much question over the proposed method as the presentation is good. However I'm worrying about the novelty and the soundness of its real-world ability comparing with other subgraph methods.

**Strengths And Weaknesses:**

Strength:
1. The author studies the K-hop message passing framework in depth, and provide several interesting theorems for regular graphs.
2. The paper is well-written and easy to follow.
Weakness:
1. The original K-hop message passing has close relationship with distance encoding proposed in Nested GNN(Zhang, 2021) and GNN-AK (Zhao, 2021), while the author ignores analyze the connections. Also the author said that they pickup a weak version of GNN-AK's result without considering distance encoding, which I think is improper especially given the close connection between distance encoding and K-hop message passing. The original GNN-AK's performance over Graph Properties and Counting Substructures is better.
2. In theorem part, stronger than 1-WL is kind of trivial to show. Also Theorem 1 is kind of trivial. Theorem 2 and Theorem 3 could be very similar to theorems in Distance Encoding paper (Li, 2020). Also analyzing expressiveness limited to regular graphs is a weakness.
3. The proposed enhanced K-hop message passing uses the information of subgraphs within kth-hop. However making using of subgraph information is well explored in Nested GNN, GNN-AK, and  Equivariant subgraph aggregation networks. Although there are some difference, the contribution is limited.
4. The experimental result in TU dataset is also questionable. The author uses two type of evaluation settings, and in the first widely used setting the author doesn't show any performance improvement. If the author want to show the improvement over the second evaluation setting, it's better to also include baselines like GIN and GIN-AK.

---

> ### Author Response · Authors · 2022-08-02
> **Author Response 1/2**
>
>
> ### **Response to reviewer qmM3**
> Thank you for your thorough and insightful comments as well as acknowledging our theoretical analysis. We reply to all the points below.
> > **Weakness 1**:
> > The original K-hop message passing has close relationship with distance encoding proposed in Nested GNN(Zhang, 2021) and GNN-AK (Zhao, 2021), while the author ignores analyze the connections.
> Also the author said that they pickup a weak version of GNN-AK's result without considering distance encoding, which I think is improper especially given the close connection between distance encoding and K-hop message passing. The original GNN-AK's performance over Graph Properties and Counting Substructures is better.
>
> **A1**:
> For the first part, as we discussed in **point 1** of the summary, the $K$-hop message passing shares some commons with distance encoding in DE-1 [1], NGNN [2], and GNN-AK [3]. However, these two methods are not exactly the same and they actually have different node-level expressive power, neither of which is strictly stronger than the other.
>
> The reasons behind comparing GNN-AK’s result without considering distance encoding are as follows: Firstly, the distance encoding in GNN-AK [3] is not their key contribution, but an additional technique they employed to further improve the results. Using distance encoding is orthogonal to GNN-AK itself. Secondly, $K$-hop GNN can also be enhanced by a similar path encoding technique (See Appendix H). For consistency and fair comparison of the pure expressive power of each model, we chose to report GNN-AK without DE in our original paper.
> <!--For the second part. the purpose for us to post the GNN-AK's result without distance encoding is that we want a fair comparsion on how much power the main contribution the proposed method bring. -->
> <!-- Secondly, The distance encoding in GNN-AK [1] is not their key contribution, it is an additional technique they employed to further improve the results. -->
> <!--The distance encoding in GNN-AK is an additional technique they used for further improvement in the experiments but not their main contribution. -->
> <!-- In fact, in all other experiments except simulation and EXP dataset, we also use novel path encoding which can be computed at no additional cost while constructing K-hop graph dataset. In Appendix G, we discuss path encoding in great depth.  -->
> Nevertheless, here we compare the full version of GNN-AK [3] with the full version of KP-GNN:
>
> **Graph properties prediction**
> | Model  | Connect. | Diameter | Radius |
> |:---:|:---:|:---:|:--:|
> |**GIN-AK+**|-2.2268|-3.7585|-5.1044|
> |**KP-GIN+**|**-4.1803**|-3.9952|-5.2206|
> |**GIN-AK+\***|-2.7513|-3.9687|-5.1846|
> |**KP-GIN+\***|-3.7229|**-4.0160**|**-5.2357**|
>
> **Counting Substructure**
> | Model  | Tri. | Tailed Tri. | Star |4-Cycle|
> |:---:|:---:|:---:|:--:|:--:|
> |**GIN-AK+**|0.0885|0.0696|0.0162|0.0668|
> |**KP-GIN+**|0.0377|0.0314|**0.0024**|0.0258|
> |**GIN-AK+\***|0.0123|0.0112|0.0150|0.0126|
> |**KP-GIN+\***|**0.0029**|**0.0027**|0.0029|**0.0082**|
>
> The \* means using additional encoding. We can see first, in graph properties prediction, the KP-GNN without encoding still outperforms the GNN-AK with distance encoding. Second, adding path encoding in KP-GNN sometimes improves the performance, especially in counting substructure where KP-GNN outperforms GNN-AK in all tasks. We include all these results in the updated version.
>
>
> > **Weakness 2**:
> In theorem part, stronger than 1-WL is kind of trivial to show. Also Theorem 1 is kind of trivial. Theorem 2 and Theorem 3 could be very similar to theorems in Distance Encoding paper (Li, 2020). Also analyzing expressiveness limited to regular graphs is a weakness.
>
> **A2**:
> 1. To further analyze the expressive power of $K$-hop message passing, we leverage the recent results in SUN [4], which characterize the boundary of the expressive power of subgraph-based GNNs like NGNN [2], GNN-AK [3], ID-GNN [5], ESAN [6]. Specifically, it implements all operations in subgraph-based GNNs using 3-IGN [7] and bounds their expressive power by 3-IGN, which is already known to be bounded by 3-WL. Here we proved that: **The $K$-hop message passing with any kernel is also bounded by 3-WL**. We do that by following the similar proof framework in SUN [4] and implementing all operations needed in $K$-hop message passing using 3-IGN.
> 2. In **point 1** of the summary, we further compare the $K$-hop message passing with distance encoding and show that **these two frameworks are not the same but have different node-level expressive power**.
> 3. In **point 2** of the summary, we further compare KP-GNN with subgraph-based GNNs by showing that: **KP-GNN is more powerful than subgraph-based GNNs with MPNN as base encoder in terms of distinguishing distance regular graphs.**
>
> By providing additional theoretical results, we further analyze the expressive power of both $K$-hop message passing and KP-GNN. We include the detailed proof and discussion in the uprated version.

---

> > ### Author Response · Authors · 2022-08-02
> > **Author Response 2/2**
> >
> >
> > > **Weakness 3**:
> > The proposed enhanced K-hop message passing uses the information of subgraphs within kth-hop. However making using of subgraph information is well explored in Nested GNN, GNN-AK, and Equivariant subgraph aggregation networks. Although there are some differences, the contribution is limited.
> >
> > **A3**: Our main contribution is to first theoretically characterize the expressive power of $K$-hop GNNs, which is not explored in any subgraph-based GNN methods like NGNN [2], GNN-AK [3]. Using $K$-hop message passing itself is very different from existing subgraph-based GNN techniques but closer to distance encoding. However, in **point 1** of the summary, we proved even these two frameworks are not the same. As shown in **point 2** of the summary, our work explores further enhancing $K$-hop GNN by peripheral subgraphs (induced subgraph by the exactly $k$th-hop neighbors), which are also different from the $k$-hop rooted subgraphs used in NGNN [2] and GNN-AK [3]. Further, according to **point 2** of the summary, KP-GNN is strictly more powerful than subgraph-based methods in terms of distinguishing distance regular graphs, which distinguishes KP-GNN from previous subgraph-based methods. Further, as in the **point 3** of the summary, the $K$-hop message passing and KP-GNN require much less space and time complexity. This further differentiates $K$-hop message passing and KP-GNN from subgraph-based GNNs.
> >
> > > **Weakness 4**:
> > The experimental result in TU dataset is also questionable. The author uses two types of evaluation settings, and in the first widely used setting the author doesn't show any performance improvement. If the author wants to show the improvement over the second evaluation setting, it's better to also include baselines like GIN and GIN-AK.
> >
> > **A4**: Thanks for the suggestion. Here we include additional experimental results on TU datasets to compare KP-GNN with GIN and GIN-AK+ [3] under the second evaluation setting. However, in the original paper of GNN-AK, the authors didn't provide detailed configuration on TU dataset experiments. For a fair comparison, we search for GIN-AK+ and GIN-AK+ with subgraphDrop, using all default parameter list in their github repository and search for the number of subgraph encoder layers (mini_layer) in {1,2,3}. For GIN, we use all settings the same as KP-GNN and search for the number of message passing layers in {2,3,4}. The results are shown below:
> >
> > | Model  | MUTAG | D&D | PTC_MR |PROTEINS|IMDB-B|
> > |:---:|:---:|:---:|:--:|:--:|:--:|
> > |**GIN**|92.8 $\pm$ 5.9|82.2 $\pm$ 3.7|65.6 $\pm$ 6.5|78.8 $\pm$ 4.1|78.1 $\pm$ 3.5|
> > |**GraphSNN**|94.70 $\pm$ 1.9|83.93 $\pm$ 2.3|70.58 $\pm$ 3.1|78.42 $\pm$ 2.7 |78.51 $\pm$ 2.3 |
> > |**GIN-AK+**|95.0 $\pm$ 6.1|OOM|74.1 $\pm$ 5.9|78.9 $\pm$ 5.4 |77.3 $\pm$ 3.1 |
> > |**KP-GIN+**|**96.1$\pm$ 4.6** |**84.0 $\pm$ 3.4**|**74.4 $\pm$ 6.5**|**80.0 $\pm$ 3.8**|**79.0 $\pm$ 2.7**|
> >
> > Here, OOM means out of memory. We can see that our proposed KP-GNN still outperforms other methods under the second setting. We have included these results in our updated version.
> >
> > References:
> > [1] Pan Li, Yanbang Wang, Hongwei Wang, and Jure Leskovec. Distance encoding–design provably more powerful gnns for structural representation learning. Advances in Neural Information Processing Systems 33 (2020): 4465-4478.
> >
> > [2] Muhan Zhang and Pan Li. Nested graph neural networks. In A. Beygelzimer, Y. Dauphin,P. Liang, and J. Wortman Vaughan, editors, Advances in Neural Information Processing Systems,2021.
> >
> > [3] Lingxiao Zhao, Wei Jin, Leman Akoglu, and Neil Shah. From stars to subgraphs: Uplifting any GNN with local structure awareness. In International Conference on Learning Representations,2022.
> >
> > [4] Fabrizio Frasca, Beatrice Bevilacqua, Michael Bronstein, and Haggai Maron. Understanding and extending subgraph gnns by rethinking their symmetries. ArXiv, abs/2206.11140, 2022.
> >
> > [5] Jiaxuan You, Jonathan Gomes-Selman, Rex Ying, and Jure Leskovec. Identity-aware graph neural networks. Proceedings of the AAAI Conference on Artificial Intelligence. Vol. 35. No. 12. 2021.
> >
> > [6] Beatrice Bevilacqua, Fabrizio Frasca, Derek Lim, Balasubramaniam Srinivasan, Chen Cai,Gopinath Balamurugan, Michael M. Bronstein, and Haggai Maron. Equivariant subgraph aggregation networks. In International Conference on Learning Representations, 2022.
> >
> > [7] Haggai Maron, Heli Ben-Hamu, Nadav Shamir, and Yaron Lipman. Invariant and equivariant graph networks. In International Conference on Learning Representations, 2019.

---

> > > ### Comment · Reviewer_qmM3 · 2022-08-08
> > > **Response to author's rebuttal**
> > >
> > > I have read all responses of the author and other reviewers' replies. I thank the author for further explanation. However my evaluation is not affected. The reason is that the main weakness is the same: not enough novelty. I would like to state it much clearly in below.
> > >
> > > 1. The analysis of general K-hop aggregation is interesting, as much old works like MixHop and K-hop-GNN use the idea directly. However there is not much interesting theorems I can find. Specifically, I personally don't care much about the regular graph. Also the upper bound with 3-WL is the main novel contribution from the new paper SUN, but NOT this paper. (It's true that you can follow their proofs, but this is not novel.)
> > >
> > > 2. The improvement from K-hop aggregation is really just encoding structural information with distance encoding. Many papers has shown that distance encoding helps adding expressivity and improves real-world performance (DE, GNN-AK, NestGNN). To be more clear, applying an existing technique (Distance Encoding, and local structural information) to an existing framework (K-hop aggregation) is **incremental**.

---

> > > > ### Author Response · Authors · 2022-08-08
> > > > **Thanks for your comments**
> > > >
> > > > Thank you again for your response and constructive comments. We understand that novelty can be subjective. Other than that, we would like to respectfully make the following clarifications.
> > > > 1. We didn't claim to propose K-hop, but rather unify existing works, theoretically study and compare their expressive power, which is unseen in previous works (they only used K-hop but did not show its improved power).
> > > > 2. Regular graphs are one of the most important classes of 1-WL-indistinguishable graphs and are of vital importance in theoretical computer science in characterizing a graph isomorphism test's power. For example, 1-WL cannot distinguish any regular graphs, while 3-WL cannot distinguish strongly regular graphs. Many previous works use regular graphs/strongly regular graphs to demonstrate an architecture's expressive power over 1-WL/3-WL. Also, the widely used EXP dataset for testing GNN's expressive power is exactly composed of regular graphs.
> > > > 3. K-hop is different from distance encoding (DE-1) as shown in our summary and Appendix C. We are **not** applying distance encoding to K-hop message passing but rather analyzing the relation and difference between the two methods. Although bothing using distance to enhance the power, their way of leveraging distance is different (DE-1 injects distance as node feature and applies normal 1-hop aggregation, while K-hop distinguishes nodes from different hops in K-hop aggregation), which results in **different expressive power** (see Appendix C).

---

### Author Response · Authors · 2022-08-02
**Summary to all reviewers and AC 1/2**

We thank all the reviewers for their constructive and insightful feedbacks. In this work, we first unify existing multi-hop GNNs into a $K$-hop GNN framework and provide theoretical analysis of its expressive power and limitation. Then we propose a new model KP-GNN to further enhance its power. We are delighted to see that our work was found well-motivated in general. Nevertheless, many reviewers pointed out our unclear contributions compared to existing works such as Distance Encoding, NGNN and GNN-AK. We address the common concerns here and address the other comments individually in our response to each reviewer. We also updated our paper based on the feedbacks (the updated part is in red).

**1. Connection and difference between $K$-hop message passing and distance encoding**

Both $K$-hop message passing with shortest path distance kernel and distance encoding [1] can be regarded as injecting distance information into nodes in original message passing framework to increase the expressive power of GNN. However, they are quite different in how to leverage the distance information. Specifically, if we only consider one layer of **$K$-hop message passing**, for every node in the graph, the distance information it injects is how many neighbors a node has at each hop. However, DE-1 (distance encoding) injects every node the distance to the target node. Then, based on the "augmented graph", DE-1 leverages **vanilla 1-hop message passing** to learn the representation of the target node. More importantly, we show that: From the node persepctive, there exists pairs of nodes from two non-isomorphic graphs that **can be distinguished by $K$-hop message passing** with shortest path distance kernel but **not by DE-1**, and **vice versa**. We prove the statement by providing two examples that can only be distinguished by one method but not the other. The first example is exactly the same pair as shown in the example 1 of Figure 1 in the paper, which cannot be distinguished by $K$-hop message passing with shortest path distance kernel. However, DE-1 is able to distinguish them by injecting distance feature. Another example is a pair of nodes in non-regular graphs but are 1-WL indistinguishable. Even with DE-1, they are still indistinguishable. However, using $K$-hop message passing, they are easy to distinguish. This further concludes that the two methods are different models with different expressive power even if they share similar ideas. That is, neither of them is strictly more expressive in node distinguishing than the other. We also notice that if we use DE-1 to label a different pair of nodes in example 2, DE-1 can still distinguish the two nodes. This means, by labeling the graph $n$ times using each node as the target node and running vanilla message passing individually on each labeled graph, DE-1 can also distinguish the two graphs. However, $K$-hop message passing only performs message passing one time over the original graph without the need to relabel and rerunning GNN iteratively. We include the detailed discussion and proof in Appendix C of our updated paper.

See the second part for the rest.

---

> ### Author Response · Authors · 2022-08-02
> **Summary to all reviewers and AC 2/2**
>
> Continue from above:
>
> **2. Connection and difference between KP-GNN and subgraph-based GNNs**
>
> The subgraph-based GNNs leverage information of subgraph to break the symmetry in the original graph. The KP-GNN also uses subgraph information to enhance the expressive power of $K$-hop message passing too. However, firstly, in KP-GNN, the message passing is performed on the **whole graph**, and **each node only has one representation**. In contrast, in subgraph-based GNNs, the message passing is performed for **each subgraph individually** and **each node can have multiple representations** depending on which subgraph it is in. Secondly, in subgraph-based GNNs, they consider the subgraph as a whole without distinguishing nodes at different hops. Instead, the $K$-hop message passing and peripheral subgraph proposed in KP-GNN can be viewed as further dividing the subgraph (ego-network) into two parts: the first part is the hierarchy of neighbors from different hops, and the second part is the connection structure of neighbors at each hop. This gives us a better point of view to design more powerful learning methods. More importantly, in our work, we proved that KP-GNN can **distinguish some distance regular graphs** in Theorem 3. However, recently, [2] points out that the **expressive power of all subgraph-based GNNs** (with 1-WL powerful model as base encoder, 1-order node selection as subgraph selection policy) is **bounded by 3-WL**. These subgraph GNNs incorporate ID-GNN [3], NGNN [4], GNN-AK [5], ESAN [6], reconstruction-GNN [7] etc. The result means that they are **unable to distinguish any distance regular graphs** without seeking for more powerful base encoder. In other words, **KP-GNN is strictly more powerful in terms of distinguishing distance regular graphs than subgraph-based GNNs**. Moreover, the $K$-hop message passing and KP-GNN can be used as the base encoder in subgraph GNNs for further improvement of expressiveness. However, it is currently unknown whether KP-GNN is strictly more powerful than subgraph-based GNNs in general and we leave it for our future work. We include the detailed discussion in Appendix F of our updated paper.
>
> **3. Time and space complexity**
>
> In our original version, we use the power of the maximum degree $d$ to characterize the time and space complexity, which makes it hard to compare with other methods. Here we analyze it from another view. Suppose graph has $n$ nodes and $m$ edges:
> (1)**The space complexity of $K$-hop message passing and KP-GNN are $O(n)$.**
> (2)**The time complexity of $K$-hop mesage passing and KP-GNN are $O(n^2)$.**
> We can see that **$K$-hop message passing and KP-GNN have the same space complexity as vanilla message passing, which is far less than $O(n^2)$ for subgraph-based GNN**. The main reason is that in $K$-hop message passing, each node only has one representation. However nodes in subgraph-based GNNs have different representations when appearing in different subgraphs. For time complexity, **KP-GNN's $O(n^2)$ time complexity also surpasses subgraph-based GNNs' $O(nm)$** significantly. These results further demonstrate the advantages of KP-GNN over subgraph-based GNNs. We include detailed discussion in Appendix G of our updated paper.
>
> Reference
>
> [1] Pan Li, Yanbang Wang, Hongwei Wang, and Jure Leskovec. Distance encoding–design provably more powerful gnns for structural representation learning. Advances in Neural Information Processing Systems 33 (2020): 4465-4478.
>
> [2] Fabrizio Frasca, Beatrice Bevilacqua, Michael Bronstein, and Haggai Maron. Understanding and extending subgraph gnns by rethinking their symmetries. ArXiv, abs/2206.11140, 2022.
>
> [3] Jiaxuan You, Jonathan Gomes-Selman, Rex Ying, and Jure Leskovec. Identity-aware graph neural networks. Proceedings of the AAAI Conference on Artificial Intelligence. Vol. 35. No. 12. 2021.
>
> [4] Muhan Zhang and Pan Li. Nested graph neural networks. In A. Beygelzimer, Y. Dauphin,P. Liang, and J. Wortman Vaughan, editors, Advances in Neural Information Processing Systems,2021.
>
> [5] Lingxiao Zhao, Wei Jin, Leman Akoglu, and Neil Shah. From stars to subgraphs: Uplifting any GNN with local structure awareness. In International Conference on Learning Representations, 2022.
>
> [6] Beatrice Bevilacqua, Fabrizio Frasca, Derek Lim, Balasubramaniam Srinivasan, Chen Cai,Gopinath Balamurugan, Michael M. Bronstein, and Haggai Maron. Equivariant subgraph aggregation networks. In International Conference on Learning Representations, 2022
>
> [7] Leonardo Cotta, Christopher Morris, and Bruno Ribeiro. Reconstruction for powerful graph representations. In A. Beygelzimer, Y. Dauphin, P. Liang, and J. Wortman Vaughan, editors, Advances in Neural Information Processing Systems, 2021

---

### Meta-Review · Area_Chair_kz6d · 2022-08-26

**Recommendation:** Accept
**Confidence:** Certain

**Metareview:**

The reviewers generally agreed that the paper is well written (with many examples), that the theoretical analysis is interesting, and that the experimental results are promising.

The reviewers raised questions regarding four main aspects of the work that were satisfactorily addressed by the authors during the rebuttal:
1. The relation between K-hop/KP-GNN message passing and other methods, such as distance encoding and subgraph-based GNNs. The authors explained the relations in a convincing manner by showing that a) KP-GNN is strictly more powerful in terms of distinguishing distance regular graphs than subgraph-based GNNs and b) by proving that neither K-hop nor distance encoding is strictly more expressive than the other.
2. The complexity of KP-GNN, which was shown to be similar to to vanilla GNN.
3. The ability of the COMBINE function to distinguish between nodes at different hops. I am convinced by the detailed explanation of the authors that this is not an issue in the setup where the message and update functions can be chosen arbitrarily. In any way, the point is not very important as the distance can be trivially incorporated.
4. The standard deviation reported in the TU dataset (Table 3) is higher than for other methods. This is true, but not a deal breaker.

Points 1 and 2 above are already addressed in the revised document. *For point 3, I ask the authors to include in their appendix a formal proof of why k is not crucial and to perform an ablation where the effect of adding k is examined.* (Concatenating k, e.g., represented as a one-hot vector, could be practically beneficial by making the learning of the message and update functions easier.)

It is my opinion that the ratings offered by the reviewers are outdated (they were not updated in light of the author rebuttal). For this reason, I decided to accept the work.

**Award:**

No

---

### Decision · Program_Chairs · 2022-09-14

Accept